# EBV renders B cells susceptible to HIV-1 in humanized mice

Donal McHugh[1],*, Renier Myburgh[2],*, Nicole Caduff[1], Michael Spohn[3], Yik Lim Kok[4,8], Christian W Keller[5], Anita Murer[1], Bithi Chatterjee[1], Julia Rühl[1], Christine Engelmann[1], Obinna Chijioke[6,7], Isaak Quast[5], Mohaned Shilaih[4], Victoria P Strouvelle[4,8], Kathrin Neumann[4], Thomas Menter[7], Stephan Dirnhofer[7], Janice KP Lam[9], Kwai F Hui[9], Simon Bredl[4], Erika Schlaepfer[4], Silvia Sorce[10], Andrea Zbinden[8], Riccarda Capaul[8], Jan D Lünemann[5], Adriano Aguzzi[10], Alan KS Chiang[9], Werner Kempf[11], Alexandra Trkola[8], Karin J Metzner[4,8], Markus G Manz[2], Adam Grundhoff[3], Roberto F Speck[4], Christian Münz[1]

**HIV and EBV are human pathogens that cause a considerable burden to worldwide health. In combination, these viruses are linked to AIDS-associated lymphomas. We found that EBV, which transforms B cells, renders them susceptible to HIV-1 infection in a CXCR4 and CD4-dependent manner in vitro and that CXCR4-tropic HIV-1 integrates into the genome of these B cells with the same molecular profile as in autologous CD4+ T cells. In addition, we established a humanized mouse model to investigate the in vivo interactions of EBV and HIV-1 upon coinfection. The respective mice that reconstitute human immune system components upon transplantation with CD34+ human hematopoietic progenitor cells could recapitulate aspects of EBV and HIV immunobiology observed in dual-infected patients. Upon coinfection of humanized mice, EBV/HIV dual-infected B cells could be detected, but were susceptible to CD8+ T-cell–mediated immune control.**

## Introduction

EBV-associated lymphomas are a considerable threat for individuals infected with the HIV type-1 (HIV-1) and constitute an AIDS–defining condition (1). With more than 90% of the adult human population being latently infected with the oncogenic γ-herpesvirus EBV (2), coinfection with EBV is the norm rather than an exception for people infected with HIV-1 (3). Nevertheless, studies exploring experimental HIV-1–EBV coinfections are lacking.

After primary infection, EBV establishes a life-long latent infection within the memory B-cell compartment with minimal viral gene expression. To gain access to this persistence reservoir, the virus relies on the expression of up to eight latency-associated genes (EBV nuclear antigens [EBNA] -1, -2, -3A-C, and -LP, and latent membrane proteins [LMP] -1 and -2) and two clusters of small noncoding RNAs (EBV-encoded RNAs [EBERs] and viral miRNAs), which promote proliferation, transformation, and confer resistance to apoptosis and immune eradication. Despite the strong B cell–transforming capacity of the virus, only few infected individuals develop EBV-associated lymphomas because of continuous restriction by EBV-specific immune responses. This immune control seems to be mainly mediated by T cells because iatrogenic T-cell–directed immune suppression leads to the increased occurrence of EBV-associated B cell lymphoproliferative diseases, which can be cured by adoptive transfer of in vitro–expanded EBV-specific T cells (4).

Chronic infection with HIV-1 results in the progressive failure of the immune system that culminates in AIDS (5). The rate of EBV association with non-Hodgkin lymphomas (NHL) in patients with AIDS is higher than with NHL in immunocompetent individuals, reaching up to 90% and 100% for systemic immunoblastic diffuse large B-cell lymphoma and central nervous system lymphoma, respectively (1, 4). The emergence of EBV-associated central nervous system lymphomas correlates with the HIV-1–induced loss of EBV-specific CD4+ T cells rather than overall CD4+ T-cell loss (6). It

[1]Viral Immunobiology, Institute of Experimental Immunology, University of Zürich, Zürich, Switzerland   [2]Department of Medical Oncology and Hematology, University and University Hospital of Zürich, Zürich, Switzerland   [3]Heinrich Pette Institute, Leibniz Institute for Experimental Virology, Hamburg, Germany   [4]Division of Infectious Diseases and Hospital Epidemiology, University Hospital of Zürich, Zürich, Switzerland   [5]Neuroinflammation, Institute of Experimental Immunology, University of Zürich, Zürich, Switzerland   [6]Cellular Immunotherapy, Institute of Experimental Immunology, University of Zürich, Zürich, Switzerland   [7]Institute of Pathology and Medical Genetics, University Hospital of Basel, Basel, Switzerland   [8]Institute of Medical Virology, University of Zürich, Zürich, Switzerland   [9]Department of Paediatrics and Adolescent Medicine, Li Ka Shing Faculty of Medicine, Queen Mary Hospital, The University of Hong Kong, Pokfulam, Hong Kong   [10]Institute of Neuropathology, University Hospital of Zurich, Zurich, Switzerland   [11]Kempf und Pfaltz Histologische Diagnostik AG, Zürich, Switzerland

Correspondence: christian.muenz@uzh.ch
Christian W Keller and Jan D Lünemann's present address is Department of Neurology, University Hospital of Münster, Münster, Germany
*Donal McHugh and Renier Myburgh contributed equally to this work

has been suggested that this loss of EBV-specific CD4[+] T cells leads to an exhausted CD8[+] T-cell population that can no longer control EBV-mediated lymphoproliferation, resulting in a progression toward NHLs (7).

Although current combined anti-retroviral therapy (cART) leads to sustained suppression of HIV RNA copy numbers in patients, poor adherence results in viral rebound because of the existence of latently HIV-1–infected cells (8). The main targets for HIV-1 replication are CD4[+] T cells, and these represent the major cell subset harbouring silent HIV-1 (8). However, HIV-1 infection is not solely restricted to CD4[+] T cells, and other cell types are also susceptible to the virus (8, 9, 10). Susceptibility to HIV-1 infection requires the co-expression of CD4 and either CCR5 (R5) or CXCR4 (X4) receptors on the cellular surface (11). In vitro, HIV-1 can infect IL-4– and/or CD40L-activated primary B cells because of the expression of CD4 and CXCR4, and infection may be enhanced in PMA-activated B cells via binding of HIV-1 in immune complexes by complement receptors CD21 and CD35 (12, 13, 14, 15). Also, HIV-1 infection of EBV-transformed and EBV-negative B cell lines has been described previously (16, 17, 18, 19, 20, 21). Although direct interactions between HIV-1 and B cells have been reported several decades ago (22, 23), there are, to our knowledge, no reports of HIV-1 replication in B cells in vivo or experimental coinfection of HIV-1 with EBV in vivo, as reviewed in reference 24.

In this study, we aimed to characterize the influence of EBV infection on HIV-1 susceptibility and possible HIV-1 integration in EBV-transformed B cells for lentiviral reservoir generation, as well as its influence on viral and host gene transcription in comparison to CD4[+] T cells in vitro. Humanized mice have been successfully used to separately investigate HIV-1 and EBV infection, pathogenesis, and immune control, and we and others could recapitulate the protective value of CD4[+] or CD8[+] T cells against EBV-mediated lymphoproliferation in these models (25, 26, 27, 28). Here, we modelled and investigated the effect of HIV-1 on EBV-specific immune control via coinfection of humanized mice. This in vivo model then allowed us to examine whether EBV-transformed primary B cells serve as target cells for HIV-1 replication upon dual virus infection.

# Results

## EBV-transformed B cells are permissive to X4-tropic but not R5-tropic HIV-1 infection in vitro

We analyzed the capacity of HIV-1 to infect EBV-transformed B cells (lymphoblastoid cell lines; LCLs) in comparison to autologous PBMCs. As expected, we observed viral replication over time, quantified by the HIV-1 p24 antigen in the supernatant, in CD19-depleted PBMCs but not CD19-purified B cells inoculated with the R5-tropic (JR-CFS) or X4-tropic (NL4-3) HIV-strains (Fig 1A). Autologous LCLs generated by in vitro EBV (B95-8 strain) infection of B cells were able to sustain X4-tropic HIV-1 (NL4-3 and HBX2) and dual-tropic HIV-1 (89.6) replication, as were LCLs derived from EBV-infected (B95.8 or M81 strain) humanized mice (Figs 1A and B and S1A and B). These results are in line with data reported previously

for several EBV-positive and EBV-negative B cell lines (14, 22, 29, 30). Despite the fact that none of the LCL donors was a carrier of the CCR5delta32 deletion (Fig S1C), LCLs were not susceptible to R5-tropic HIV-1 strains (YU-2 and JR-CSF), evidenced by the lack of p24 in the culture supernatant and the absence of HIV-1 mRNA transcript splice variants (Figs 1B and S1B). In line with the selective susceptibility of LCLs to X4-tropic HIV-1 infection, we found that a large fraction of the LCLs expressed CD4 on their surface and retained expression of CXCR4 at lower levels upon EBV transformation, whereas transcript levels of CCR5 were very low in selectively sequenced LCLs as previously reported (Figs 1C and S1D) (31, 32, 33). The combined surface expression of CD4 and CXCR4 of individual LCLs correlated strongly with HIV-1 replication over time in vitro (Fig 1D). Furthermore, upon FACS, pure populations of CD4[+] and CD4[−] LCLs either lost or gained CD4 expression, respectively, within days of in vitro culture, indicating that the CD4[+] cells are unlikely derived from a clonal population. The resultant CD4[high] LCL subcultures were more susceptible to X4-tropic HIV-1 replication compared with the autologous CD4[low] subcultures (Fig S1E and F). In addition, X4-tropic HIV-1 replication was efficiently suppressed by anti-retroviral treatment (ART: Efavirenz & AZT) in vitro in both donor LCLs and autologous CD4[+] T cells (Fig S1G and H). To assess if EBV infection can induce CD4 expression in humans, we investigated PBMCs from healthy blood donors, patients with infectious mononucleosis, that is, the primary symptomatic infection with EBV, or EBV[+] B cell posttransplant lymphoproliferative disorders (PTLDs) and found that patients with EBV-associated diseases had more frequent CD4[+] B cells than controls (Fig 1E and Table S1). Moreover, in a separate cohort of 10 EBV[+] B cell PTLDs, we found evidence of CD4 surface expression on EBV-infected (EBER[+]) cells in two patients (Fig 1F and Table S2). These previously described PTLDs were also positive for CXCR4 (34). In summary, EBV transformation can induce B-cell susceptibility to HIV-1 infection in vitro in a CD4- and CXCR4-dependent manner, similar to IL-4 and/or CD40L stimulation of B cells as shown previously (12, 14) and CD4 can be detected on peripheral blood B cells and lymphoma cells in EBV-associated diseases.

## X4-tropic HIV-1 host genome integration profile in LCLs is similar to autologous T cells

HIV-1 proviral integration into the host cell genome is an essential part of the retroviral life cycle and lentiviral reservoir generation (35). Therefore, we investigated HIV-1 integration into the host genome of LCLs and whether integration profiles differ between LCLs and CD4[+] T cells. Purified CD4[+] T cells and autologous LCLs enriched for CD4 expression by FACS were infected with X4-tropic HIV-1 for 2 d. HIV-1 integration sites were amplified with a non-restrictive linear amplification-mediated PCR (nrLAM-PCR) (36, 37), sequenced, and reads were mapped using the Integration Site Analysis Pipeline (37). We found that HIV-1 integrates into the genome of LCLs with an overrepresentation of sites within transcriptional units similar to CD4[+] T cells (37, 38, 39) (Fig 2A and Table S3) and monocytes/macrophages (37, 40). We observed the HIV-1 signature weakly conserved palindromic nucleotide sequence (GT(A/T)AC) in the host genome upstream of HIV-1 integration in both the CD4[+] T cells and the LCLs (Fig 2B), as previously described

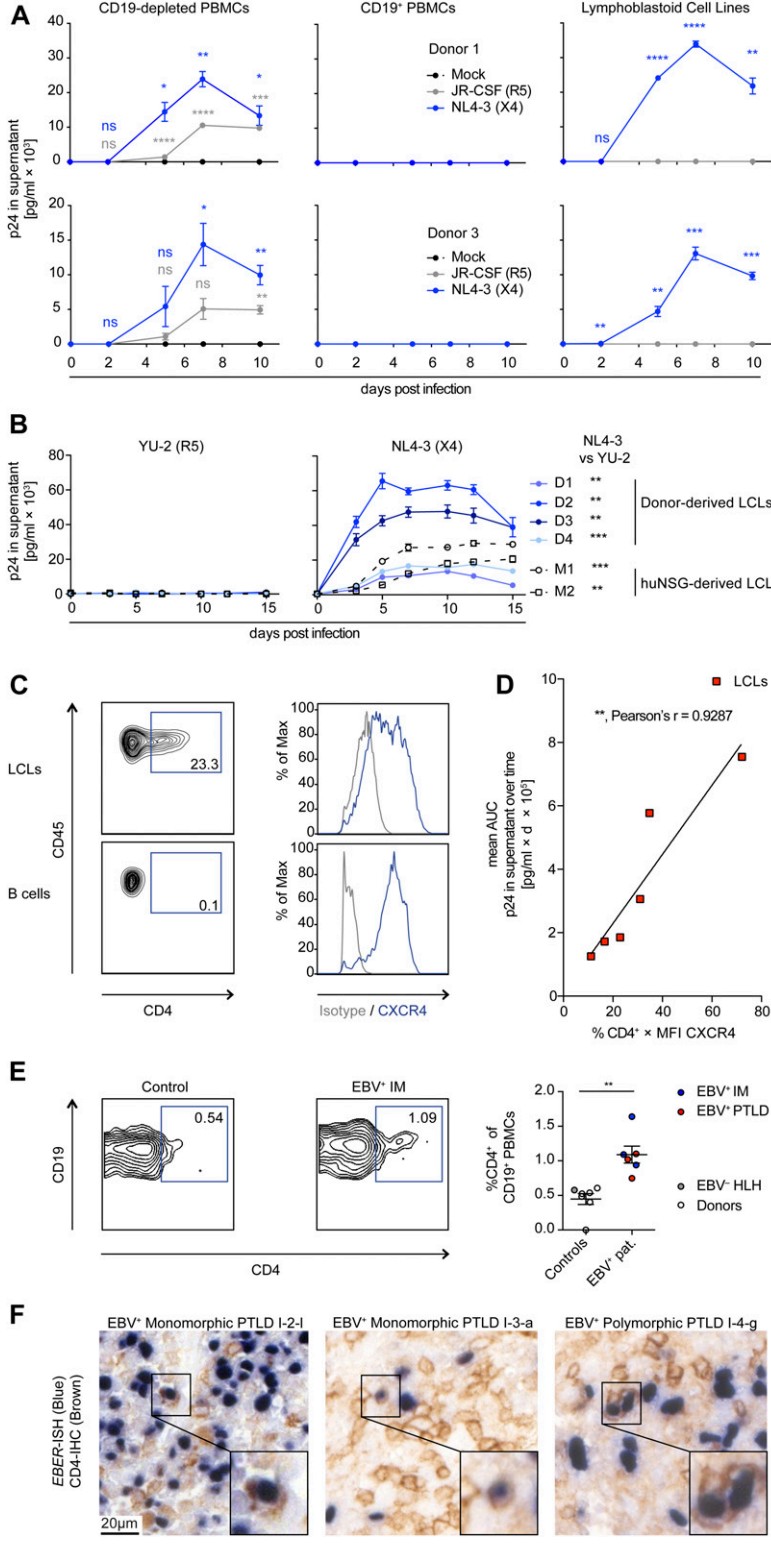

**Figure 1. X4-tropic HIV-1 replicates in EBV-transformed B cells in vitro.**
**(A)** Quantification of p24 via ELISA in supernatants collected over 10 d from in vitro HIV-1–infected CD19⁺ B-cell–depleted PBMCs, purified CD19⁺ B cells and lymphoblastoid cell lines (LCLs) from two donors. Cells were either mock-infected, infected with JR-CSF (R5-tropic HIV-1), or NL4-3 (X4-tropic HIV-1). Data from two donors are depicted. Adjusted *P*-value summaries for comparison of NL4-3 and JR-CSF versus Mock are indicated in blue and grey, respectively, from two-tailed unpaired *t* tests, corrected by the Holm–Sidak method. **(B)** Quantification of p24 in supernatants collected over 15 d from HIV-1–infected LCLs derived from four donors and two EBV-infected humanized mice reconstituted with fetal liver-derived CD34⁺ cells. Cells were infected with YU-2 (R5-tropic HIV-1) or NL4-3. The area under the curve (AUC, p24 in the supernatant versus time postinfection) was compared via two-tailed unpaired *t* test with Welch's correction. **(C)** Representative flow cytometry plots and histograms of LCLs and autologous purified B cells stained for CD45, CD4, and CXCR4. **(D)** Correlation of NL4-3 HIV-1 replication in different LCLs with the level of CXCR4 and CD4 surface expression before infection. HIV-1 replication was approximated via analyzing the AUC for each LCL as shown in (B). Relative surface expression of HIV entry receptors for each LCL was approximated by multiplying the frequency of CD4⁺ cells with the median fluorescence intensity of CXCR4 as determined by flow cytometry immune phenotyping. Correlation, **$P$ = 0.0074, Pearson's r = 0.9287. **(E)** Representative flow cytometry immune phenotyping logarithmic contour plots and quantification of CD4 surface expression on CD19⁺ B cells from controls (healthy blood donors and 1 EBV⁻ hemophagocytic lymphohistiocytosis [HLH] patient) and EBV⁺ infectious mononucleosis or EBV⁺ posttransplant lymphoproliferative disorder (PTLDs) patients. Events were pre-gated on single cells/lymphocytes/CD3⁻/CD19⁺ cells. **$P$ = 0.001 (Mann–Whitney test). **(F)** Dual *EBER* in situ hybridization (blue) CD4 immunohistochemistry (IHC, brown) on tissue microarrays of two cases of EBV⁺ PTLD. I-2-l and I-3-a are separate sections from the same PTLD. Scale bar: 20 μm. Inserts are a 2× magnification of a section of the main image. In (A, B, C), data are represented as mean ± SEM and were performed in triplicate. Results representative of four donors. *P*-values are reported as ns > 0.05, * < 0.05, ** < 0.01, *** < 0.001, **** < 0.0001.

for T cells (37, 41, 42, 43). In line with previous findings (39, 42), we did not observe a preference for the orientation of the HIV-1 provirus relative to the transcriptional direction of host genes in CD4⁺ T cells and LCLs (Fig 2C). To correlate HIV-1 proviral integration with gene

transcription in LCLs and CD4⁺ T cells, RNA sequencing was performed with LCLs and autologous CD4⁺ T cells 2 d after infection with X4-tropic HIV-1 or mock infection. Upon ranking host gene expression in mock-infected CD4⁺ T cells and autologous LCLs

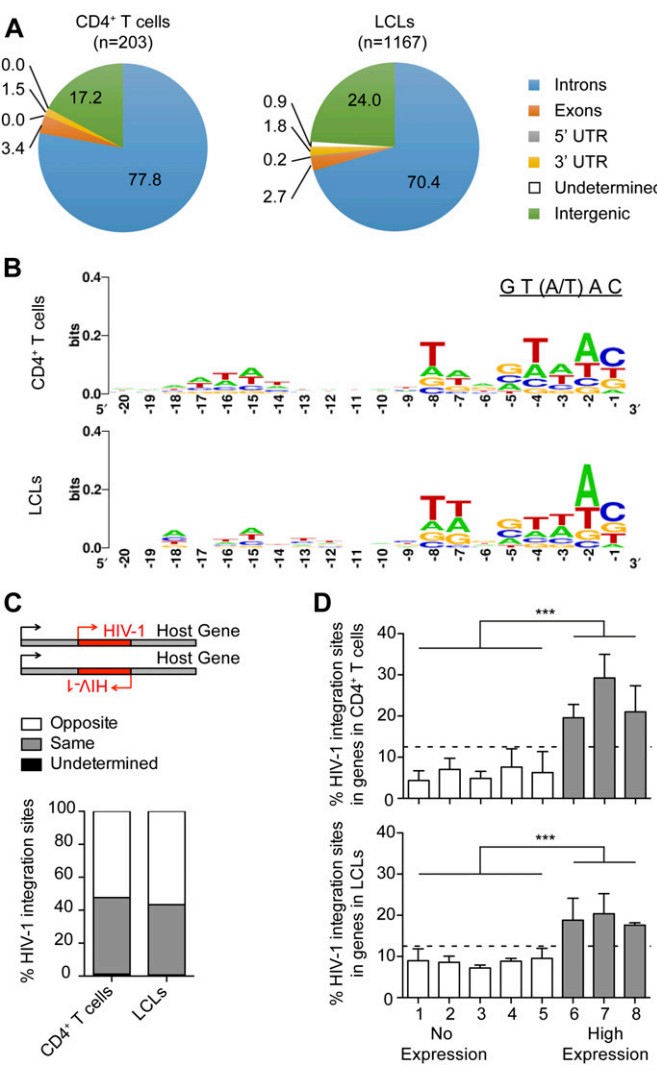

**Figure 2. X4-tropic HIV-1 host genome integration profile in lymphoblastoid cell lines (LCLs) and autologous T cells.**
**(A)** Distribution of HIV-1 integration sites in the host genome comparing CD4+ T cells and autologous LCLs 2 d post in vitro infection with NL4-3 (n = number of integration sites). **(B)** Nucleotide sequence consensus upstream of HIV-1 5′LTR in the host genome. **(C)** Transcription orientation of intragenic HIV-1 relative to the host gene. HIV-1 integration events in loci of transcript variants or more than one gene with different features of interest are classified as undetermined. **(D)** Frequency of HIV-1 integrations in genes of autologous donor CD4+ T cells and LCLs. Host genes in noninfected LCLs and CD4+ T cells were ranked from non-expressed to highly expressed genes based on reads per kilobase of RNA transcript per million mapped reads (RPKM). The host genes were grouped into eight bins of equal size based on RPKM. Only the grey bins contain genes with an RPKM value greater than 0. The relative distribution of the total HIV-1 integration sites among the eight bins is summarized for the CD4+ T cells and LCLs. ***$P < 0.0001$ (Chi-squared test). **(A, B, C, D)** Data derived from three donors: two LCLs and three CD4+ T cells each with two separate HIV-1 or mock infections; two of the T cells were autologous to the investigated LCLs.

based on reads per kilobase of RNA transcript per million mapped reads (RPKM), we grouped these genes into eight bins of equal sizes and observed that integration sites were overrepresented in bins containing actively transcribed genes (Fig 2D). These data indicate that HIV-1 favors integration in actively transcribed genes in both LCLs and CD4+ T cells confirming previous reports for CD4+ T-cell

infection (38, 44). We also observed a trend toward reduced integration in the most transcriptionally active set of genes. Because of their more central location in the nucleus, highly transcribed genes may be less accessible for viral integration, which often occurs close to the nuclear pore (44). HIV-1 integration in LCLs was found to be enriched in genes for endosomal trafficking (gene ontology [GO] term: 0006895, Golgi to endosome transport, 9.89-fold enriched, false discovery rate [FDR] = 0.003; GO term: 0032456, endocytic recycling, 7.32-fold enriched, FDR = 0.011), whereas HIV-1 integration in CD4+ T cells was primarily enriched for genes of the choline transport (GO term: 0015871, 51.24-fold enriched, FDR = 0.019; Table S2). However, in both LCLs and T cells, HIV-1 proviruses also targeted genes that regulate mitotic cell cycle phase transition (GO term: 1901990; LCLs: 2.15-fold enriched, FDR = 0.013, T cells: 3.69, FDR = 0.048), indicating frequent integration into cell cycle–associated genes. These genes are actively transcribed in LCLs as well as activated T cells and are thus likely more accessible to HIV-1 integration (Table S4). It remains, however, unclear if these integration events alter the expression of the respective host genes and thereby affect lymphocyte growth behaviour. No HIV-1 integrations into the EBV genome were detected. In summary, the integration profiles of X4-tropic HIV-1 in the host genomes of CD4+ T cells and LCLs follow the same pattern.

## CD8+ T cells expand but do not control EBV during EBV/HIV dual infection of humanized mice

Because in vitro infection of cells cannot fully recapitulate the effects of EBV plus HIV-1 dual infection with respect to the induced immune responses, we investigated EBV–HIV-1 interactions in an in vivo model of infection and immune control. NOD-*scid* $γ_c^{-/-}$ Tg(HLA-A2) (NSG-A2) mice with human immune system components (humanized mice) reconstituted from CD34+ hematopoietic progenitor cells (HPCs) were infected with EBV (B95-8 (45)) and 1 wk later with HIV-1 (NL4-3). The experiment was terminated at 4 wk post-EBV infection because of the considerable weight loss of EBV/HIV dual-infected mice (Figs 3A and S2A). Conventional H&E and immuno-histochemistry (IHC) staining of splenic sections from dual-infected animals revealed the presence of tumor-like lesions containing CD20+ B cells with a high frequency of EBNA2-positive cells and similar CD8+ T-cell infiltration compared with EBV-infected mice (Fig 3B). No significant differences were detected in the relative expression of latent and lytic EBV transcripts in non-tumorous splenic B cells isolated from either EBV or EBV/HIV dual-infected mice (Fig S2B). We observed a small but significant loss of CD4+ T cells in the blood of HIV-infected mice, an expansion of both CD8+ and CD4+ T cells in EBV-infected mice, and a similar expansion of CD8+ but not CD4+ T cells in EBV/HIV dual-infected mice (Fig 3C). Particularly, the memory CD8+ T cells were increased, as determined in the spleen (Fig S2C). Further analysis of splenic CD8+ T cells revealed a trend towards an activated, but exhausted phenotype in EBV/HIV dual-infected mice with enhanced surface expression of HLA-DR, PD1, and Tim3 (Fig S2D). Although serum cytokine levels of IFNγ, TNFα, IL-6, and IL-10 were similarly high in EBV and EBV/HIV–infected mice (Fig S2E), the ex vivo EBV-specific IFNγ release from CD19-depleted splenocytes challenged with autologous LCLs was lower in dual-infected animals (Fig 3D). These findings mirror data comparing

IFNγ release of EBV-specific CD8[+] T cells derived from HIV-1–infected individuals that either remained long-term asymptomatic or progressed to AIDS due to opportunistic infections (46). As such, despite a comparable CD8[+] T-cell expansion in EBV and EBV/HIV dual-infected mice, T cells from a dual-infected environment did not seem to react to EBV-infected cells as efficiently. Furthermore, macroscopically visible tumors in the spleen, pancreas, and liver were more frequent in EBV/HIV dual-infected mice than EBV single-infected mice and coincided with a trend towards higher EBV burden in the spleen (Fig 3E and F). Moreover, CD8[+] T-cell depletion via OKT8 treatment (Fig S2F and G) only exacerbated the EBV disease burden in EBV single-infected mice, whereas in EBV/HIV dual-infected animals, OKT8 treatment did not result in a further increase in tumor burden or EBV viral load (Fig 3E and F). In contrast to a previous study in which HIV-1–specific immune control was compromised upon antibody-mediated CD8[+] T-cell depletion at 2 and 5–7 wk post-HIV-1 infection (47), in our hands, CD8 depletion 1 wk post-HIV-1 infection did not lead to measurable differences in serum HIV-1 copy numbers at 2 wk post-depletion (Fig 3G). In summary, these data indicate that CD8[+] T cells expand and can control EBV during EBV-single infection and despite expanding in EBV/HIV dual-infected mice, they seem to partially lose EBV-specific immune control leading to enhanced EBV-disease progression.

### B cells from EBV/HIV dual-infected humanized mice can transmit HIV-1 infection to autologous humanized mice

Based on the ability of HIV-1 to infect LCLs in vitro (Fig 1), we aimed to determine whether HIV-1 infects EBV-infected B cells in humanized mice. Co-IHC detecting the B-cell lineage specific transcription factor PAX5 and the HIV-1 p24 antigen on splenic sections revealed the presence of PAX5[+] cells positive for cytoplasmic p24 in EBNA2[+] regions of the spleens of dual-infected animals when CD8[+] T cells were depleted (Fig 4A). EBNA2[+]/p24[+] cells observed in dual-infected animals were present at a significantly higher frequency in CD8-depleted mice and comprised a larger fraction of all the HIV-infected cells within tumors than non-tumorous spleen tissue (Fig 4B and C). In line with these findings, multiple spliced HIV-1 RNA transcripts were detected more frequently in purified B-cell fractions from dual-infected CD8-depleted animals (Fig S3A). We aimed to determine whether in vivo EBV/HIV coinfected B cells actively replicate HIV-1 upon transfer to a HIV-1 naïve host. To this end, humanized mice were infected with EBV and 1 wk later with X4-tropic HIV-1, followed by treatment with either CD8[+] T-cell–depleting antibody or PBS at week 2 and 4 post-EBV infection (Fig 4D). From these dual-infected mice, splenic CD19[+] B-cell fractions were purified by repeating magnetically activated cell sorting twice, to ensure low T-cell contamination (Fig S3B). These B-cell fractions or an equal number of CD19-depleted splenocytes obtained from dual-infected non–CD8-depleted or dual-infected CD8-depleted donor mice were transferred to individual humanized littermates reconstituted with the same human fetal liver (HFL)-derived CD34[+] HPCs. Emergence of EBV and HIV was monitored in the blood and plasma of recipient animals for 6–7 wk and, upon euthanizing, in the spleen via IHC for EBNA2 and p24 (Fig 4D). As expected, most of the mice that received the CD4[+] T-cell containing CD19-depleted fractions from both donor

groups developed HIV-1 plasma viral loads (Fig 4E). Supporting our finding that B cells can harbour HIV in EBV/HIV dual-infected humanized mice, some recipients of CD19[+] B-cell fractions from dual-infected CD8-depleted mice developed detectable HIV-1 RNA loads in the plasma, whereas recipients of B cells from dual-infected non–CD8-depleted mice did not (Fig 4E).

Next, we investigated the potential of EBV[+] B cells to function as a reservoir for latent HIV-1 infection upon cART treatment in vivo in the absence of T-cell immune control. We infected LCLs derived from humanized mice with HIV-1 and transferred these cells into NSG mice, which we then either treated with cART via the drinking water or normal drinking water for 2 wk, followed by another 2 wk of normal drinking water. To control for effective HIV-1 suppression, humanized mice infected with HIV-1 for 4 wk were subjected to the same 2-wk cART treatment (Fig S3C). We found evidence of HIV-1 persistence in cART-treated mice either by IHC for p24 in the tumor tissue, HIV-1 DNA integration, spliced HIV-1 RNA, or quantification of HIV-1 genomes in the plasma (Fig S3D). In summary, these data indicate that during conditions of reduced CD8[+] T-cell–mediated immune surveillance, HIV can readily infect EBV-infected B cells, and these cells can either contribute to viremia or potentially serve as a reservoir for HIV-1 during cART treatment.

### HIV-1–infected LCLs are more susceptible to elimination by autologous T-cell clones in vitro

To investigate the effect of HIV infection on the host transcriptome in LCLs and CD4[+] T cells, the aforementioned RNA sequencing was mined for differentially expressed host and viral genes. This gene expression analysis revealed differential transcription of HIV genes in the two cell types. The nef gene was highly expressed in CD4[+] T cells compared with other HIV genes and in contrast to the HIV gene expression pattern found in LCLs (Fig 5A). Conversely, hierarchical clustering analysis based on the EBV transcriptome did not show clustering of HIV-1–infected and noninfected LCLs, respectively (Fig S4). In general, HIV-1 affected the host transcriptome of LCLs less than that of CD4[+] T cells. However, gene set enrichment analysis detected significant disparate expression of genes included in GO sets related to antigen processing and/or presentation of peptide antigen via MHC class I between HIV-1–infected T cells and LCLs relative to mock-infected controls (Fig 5B and Table S5). Specifically, genes related to "antigen processing and presentation of peptide antigen via MHC class I" were significantly down-regulated in HIV-1–infected CD4[+] T cells and up-regulated in HIV-1–infected LCLs upon GO enrichment analysis. This GO set is composed of 90 genes including genes encoding for TAP1, calreticulin, and subunits of the immunoproteasome, including Mecl-1, the large multifunctional peptidases 2 and 7 (LMP2 and 7), and the proteasome activator components Pa28α and β (Fig S5A). Based on these changes in genes involved in enhanced antigen processing and presentation via MHC class I in LCLs, we investigated whether HIV-1 infection renders LCLs more susceptible to elimination by CD8[+] T cells. Five EBV-specific CD8[+] T-cell clones isolated from four EBV[+] donors with specific reactivity toward endogenously processed EBV latent antigens (Fig 5C) were cultured with autologous or HLA class I–matched X4-tropic HIV-1–infected LCLs. After 18 h of co-culture with the EBV-specific CD8[+] T-cell clones, the fraction of p24[+] LCLs in the

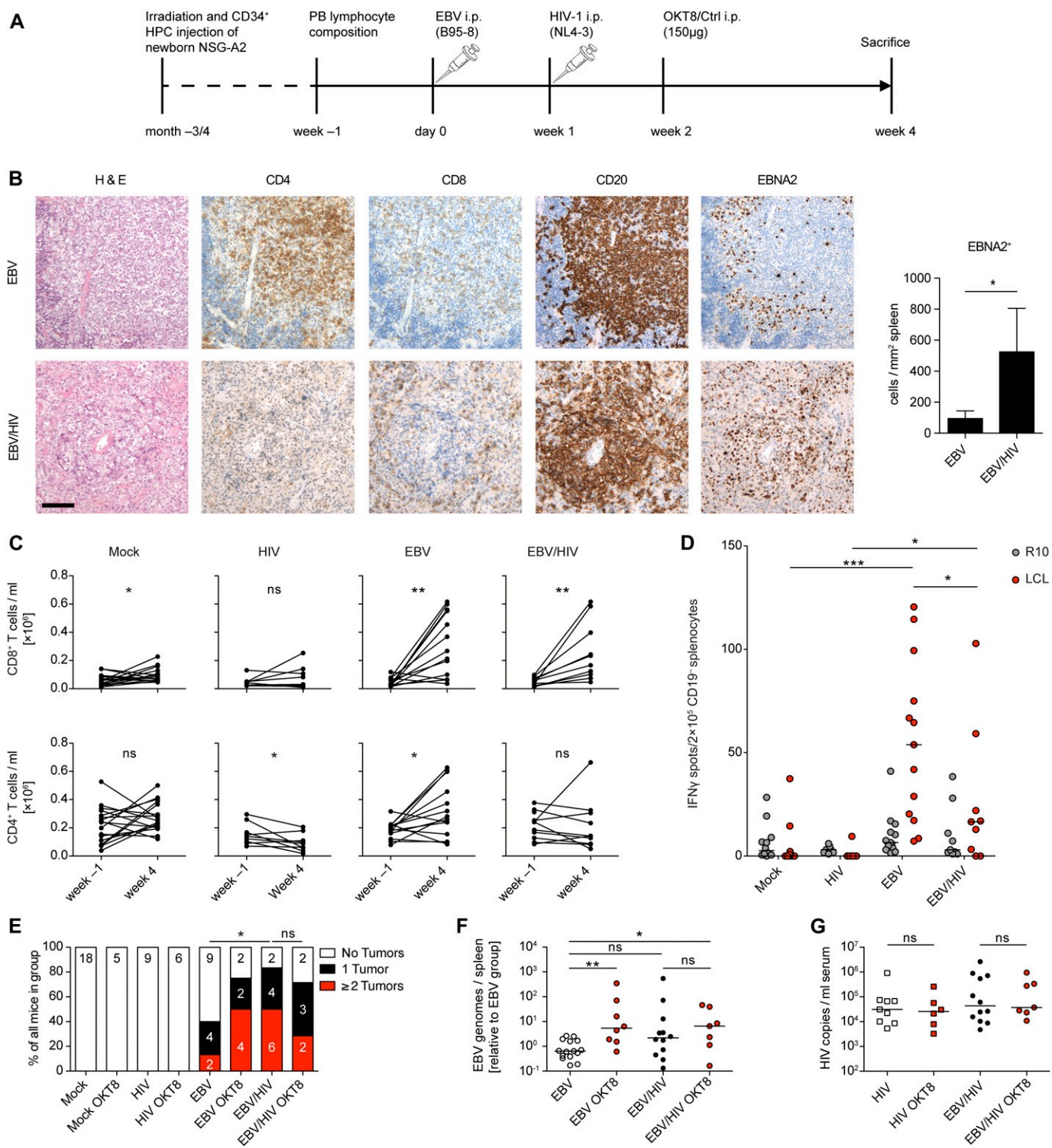

**Figure 3. EBV/HIV dual-infection of humanized mice.**
**(A)** Experimental setup: newborn NSG-A2 mice were irradiated and received intrahepatic injection of human CD34[+] hematopoietic progenitor cells. Before infection, baseline blood lymphocyte composition was determined by flow cytometry and mice were grouped. At day 0, mice were infected i.p. with EBV or PBS and 1 wk later with HIV-1 or PBS. At week 2 post-infection, mice received a single injection of a CD8-depleting monoclonal antibody (OKT8) or an isotype control antibody. In one experiment, mice were left untreated. Mice were euthanized 4 wk post-EBV infection. **(B)** Representative histological panel of formalin-fixed, paraffin-embedded spleen sections from EBV and EBV/HIV–infected huNSG-A2 mice depicting H&E, CD4, CD8, CD20 and EBNA2 IHC stainings. Scale bar: 100 μm. Quantification of EBNA2[+] cells per square millimeter with n = 7 and 5 mice, respectively. *P = 0.030 (Mann–Whitney test, MWT). **(C)** Number of CD8[+] and CD4[+] T cells per ml blood at week −1 and at week 4. Mock; CD8[+] *P = 0.049. HIV; CD4[+] *P = 0.042. EBV; CD8[+] **P = 0.001 and CD4[+] *P = 0.021. EBV/HIV; CD8[+] **P = 0.007 (two-tailed paired t test). **(D)** EBV-specific T cell ELISpot assay. Mean IFNγ release was quantified upon co-culture of CD19-depleted splenocytes derived from humanized mice of the indicated groups with autologous EBV-transformed

total LCL population decreased, indicating a more efficient elimination of HIV-1–infected cells than the non–HIV-1–infected LCLs (Fig 5D). Interestingly, cell surface expression levels of HLA class I were down-regulated in both HIV-1–infected LCLs and autologous HIV-1–infected CD4$^+$ T cells despite lower relative abundance of nef in LCLs, which has been described to down-regulate MHC class I surface expression (48) (Fig S5B). These data indicate that HIV-1–infected LCLs are highly sensitive to CD8$^+$ T-cell immune control despite global HLA class I down-regulation. HIV-1–infected LCLs were also clearly susceptible to HIV-specific CD8$^+$ T-cell clones derived from three individual donors (Fig S5C), whereas influenza-specific CD8$^+$ T cells did not react to HIV$^+$ LCLs preferentially (Fig S5D). Thus, EBV/HIV coinfected B cells appear to be more efficiently targeted by CD8$^+$ T cells specific for HIV-1 and EBV in comparison to EBV single-infected B cells. The enhanced vulnerability of EBV-infected B cells upon HIV-1 coinfection may explain why these coinfected cells were preferentially observed upon CD8 depletion in vivo, and point toward HIV-1 superinfected EBV-transformed B cells being efficiently controlled by CD8$^+$ T cells, even if their activity to protect from EBV-induced lymphomas is compromised by loss of CD4$^+$ T-cell help due to HIV-1 infection.

## Discussion

There is increasing evidence that cell types other than CD4$^+$ memory T cells have the potential to constitute a fraction of the overall HIV-1 latency reservoir (8, 10). To date, efforts to identify and phenotype cells that make up the HIV-1 reservoir have located HIV-1 provirus in tissue-resident macrophages during cART treatment of a humanized mouse model, in which only human myeloid and B cells develop from a human CD34$^+$ HPC graft (49). Furthermore, HPCs have been reported to sustain HIV-1 infection both in vitro and in vivo in patients and in a humanized mouse model (50, 51). We believe that activated B cells could serve as HIV-1 target cells, especially upon infection with the ubiquitous γ-herpesvirus EBV. As such, HIV-1 could potentially directly contribute to malignant transformation or reside latently in the long-lived EBV-transformed memory B cells hidden from the immune system and thus constitute a potential additional long-lived reservoir for HIV-1 latency. Indeed, we found a similar integration profile of X4-tropic HIV-1 in autologous LCLs and CD4$^+$ T cells in vitro. Furthermore, we provide evidence that EBV-transformed cells may harbour HIV-1 in vivo after ART treatment, during which no peripheral HIV-1 RNA load can be detected. Dual-infection, however, is likely a rare event as it pertains solely to X4-tropic viruses which occur in only 50% of patients at late-stage disease (52) and only a fraction of the EBV-infected B cells express the requisite surface receptors CD4

and CXCR4. Furthermore, we found that dual-infected cells are highly susceptible to immune control via CD8$^+$ T cells in vitro and may only contribute to HIV-1 viremia when CD8$^+$ T-cell function is severely impaired. Because we found that the conditions for X4-tropic HIV-1 susceptibility were met in patients with acute EBV infection or EBV-related lymphomas, future studies could investigate whether HIV infection of EBV$^+$ B cells occurs in people living with HIV, especially during cART.

We found no evidence that HIV-1 infection perturbs EBV gene expression directly in B cells derived from EBV/HIV dual-infected animals. Similarly, upon in vitro infection of LCLs with HIV-1, the abundance of different EBV transcripts did not indicate a shift in the latency program or an induction of the EBV lytic cycle. This is in contrast to the enhanced EBV lytic gene expression and plasma cell differentiation observed upon EBV/KSHV dual-infection (53). We did observe differential HIV-1 gene expression upon HIV-1 infection in LCLs compared with autologous HIV-1–infected CD4$^+$ T cells, which may be explained by different HIV-1 replication dynamics in LCLs compared with T cells (54, 55, 56). This was accompanied by discrete changes in the host transcriptome, whereby genes related to MHC-I–mediated antigen presentation were enriched in HIV-1–infected compared with noninfected LCLs in contrast to HIV-1–infected versus noninfected T cells. Thus, dual-infections of EBV-infected B cells with either HIV-1 or KSHV have a different impact on both EBV and host gene transcription.

Infection of humanized mice with a dose of EBV as used in this study leads to detectable EBV DNA levels, relatively asymptomatic EBV persistence with the development of few EBV-driven tumors and an EBV-specific T-cell response. CD8$^+$ T cells expand and infiltrate EBV-associated tumors to a similar degree upon EBV single- and EBV/HIV dual-infection. However, the additional burden of HIV-1 infection resulted in less CD4$^+$ T-cell infiltration, an increase in EBV-associated tumors and a trend towards an exhausted CD8$^+$ T-cell phenotype. The latter observation has also been made in humanized mouse infections with HIV-1 over longer experimental periods (57). Furthermore, we observed a reduced EBV-specific T-cell response and CD8 depletion did not lead to a higher EBV-associated tumor burden in dual-infected animals. These results, modelled within a very short time frame, mirror the compromised EBV-specific immune control preceding EBV-associated lymphomagenesis that has been observed in HIV-1–infected patient cohorts (6, 7). As such, we find this animal system informative for investigating these human lymphotropic viruses because it recapitulates features of human EBV-specific CD8$^+$ T-cell dysfunction associated with HIV-1.

However, some CD8$^+$ T-cell function was retained upon dual infection. When CD8$^+$ T cells were depleted in EBV/HIV dual-infected mice, an increased number of EBNA2/p24 double-positive cells could be found in the spleen compared with non-depleted dual-infected mice. This points to a preferential immune control

---

B cells (lymphoblastoid cell lines or LCLs) or medium (R10). Mock versus EBV ***$P$ < 0.001; HIV versus EBV/HIV *$P$ = 0.013; EBV versus EBV/HIV *$P$ = 0.045 (MWT). **(E)** Presence of macroscopically visible EBV-associated tumors in animals from the indicated experimental groups. *$P$ = 0.016 (MWT for tumor score). **(F)** Total splenic EBV DNA burden was determined for each mouse by qPCR for EBV BamHI W fragment and plotted relative to EBV-infected animals. EBV versus EBV OKT8 treated *$P$ = 0.005; EBV versus EBV/HIV $P$ = 0.060; EBV versus EBV/HIV OKT8 treated **$P$ = 0.035; EBV/HIV versus EBV/HIV OKT8 treated $P$ = 0.583 (MWT). **(G)** Serum HIV RNA copy numbers were determined by RT-qPCR at week 4 (MWT). **(B, C, D, E, F, G)** Represents pooled data from three experiments. **(F, G)** Individual values for each mouse and median are depicted.

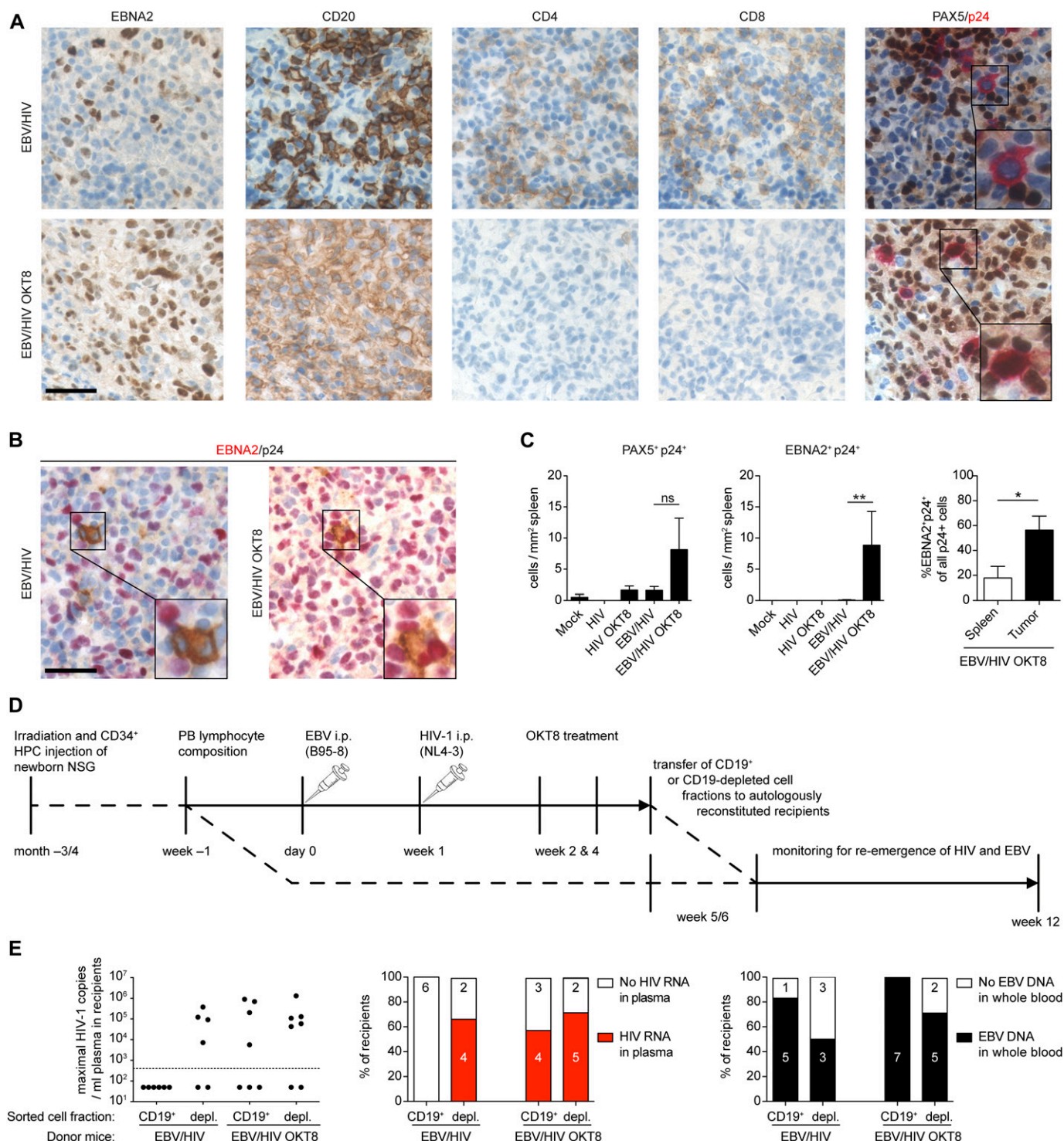

**Figure 4. EBV/HIV dual-infected humanized mice replicate HIV-1 in B cells.**
**(A)** Representative IHC stainings of EBNA2, CD20, CD4, CD8, and co-IHC for PAX5 (brown, nuclear stain) and p24 (red, cytoplasmic stain) of formalin-fixed, paraffin-embedded spleen sections from EBV/HIV–infected and EBV/HIV–infected OKT8-treated mice (EBV/HIV OKT8). **(B)** Representative co-immunohistochemistry stainings for EBNA2 (red, nuclear staining) and p24 (brown, cytoplasmic staining). **(A, B)** Scale bar: 40 μm. Inserts are a 2× magnification of a section of the main image. **(C)** Quantification of PAX5$^+$/p24$^+$ (left) and EBNA2$^+$/p24$^+$ (mid) cells in the spleen from the indicated experimental groups and EBNA2$^+$/p24$^+$ as % of total p24$^+$ (right) cells in non-tumorous spleen and tumor tissue within the EBV/HIV OKT8 group. N = respectively; Mock: 4 & 5, HIV: 6 & 3, HIV OKT8: 7 & 5, EBV/HIV: 14 & 12, EBV/HIV OKT8: 15 & 14, "Spleen": 5, "Tumor": 4. PAX5$^+$/p24$^+$ $P$ = 0.112; EBNA2$^+$/p24$^+$: **$P$ = 0.006, Spleen versus Tumor: *$P$ = 0.032 (Mann–Whitney test). **(D)** Experimental setup for cell fraction transfer assay: Newborn NSG mice were irradiated and transplanted with fetal liver derived CD34$^+$ hematopoietic progenitor cells. Donor mice were infected with EBV and 1 wk later with X4-tropic HIV-1. At week 2 and 4, donor mice either received the CD8-depleting antibody (OKT8) treatment or PBS. At week 5 or 6, the donors were sacrificed, and

of HIV-1 coinfected over EBV single-infected B cells by CD8[+] T cells. Indeed, latently EBV-infected B cells minimize their recognition by CD8[+] T cells with low latent EBV antigen expression, resulting in less than one peptide presented on MHC class I molecules per cell for some T-cell epitopes of EBNA3C, the antigen which our HLA-B*4402-restricted CD8[+] T-cell clone recognized in this study ([58]). This often allows for only 30% maximal specific lysis by CD8[+] T-cell clones, such as the HLA-B*0850-restricted EBNA3A-specific T-cell clone used in this study ([59]). Finally, EBNA1, recognized by two HLA-B*3501-restricted CD8[+] T-cell clones in this study, limits its own translation and proteasomal degradation through β-sheet formation with its glycine–alanine repeat domain ([60], [61]). Upon super-infection with replication competent HIV-1, many HIV-1 proteins are expressed, and it is conceivable that a professional antigen-presenting B cell may protect itself from this high protein production by inducing the immunoproteasome by default. As such, the observed transcriptional up-regulation of antigen processing for MHC class I presentation seems to enhance latent EBV antigen recognition by CD8[+] T cells. The resulting immunoproteasome induction has indeed been described to increase the process of cytosolic proteolysis for optimal MHC class I ligand generation ([62], [63], [64], [65], [66]). This suggests that HIV-1 coinfection increases antigen processing of viral antigens, rendering the coinfected B cells susceptible to immune control in part because of improved EBV-specific recognition and additionally via HIV-1–specific CD8[+] T-cell recognition. Thus, HIV-1 can use EBV-infected B cells for integration, replication, and transfer in vivo, but in turn, this target cell for HIV-1 is efficiently controlled by CD8[+] T cells.

Humanized mice have the ability to recapitulate certain aspects of the human immune system in vivo. However, a number of limitations in immune function should be mentioned at this point, which may preclude direct transfer of these results to human infections (reviewed in reference [25]). Importantly, reconstituted human immune system components show similarities to cord blood immune cells. Although cell-mediated immune responses can be mounted, the magnitude of these responses may be lower than in human adults, and isotype switched antibody responses are only rarely observed and steady state levels of IgG are a 1,000-fold lower than in adult human serum. As such, HIV-1 binding on the surface of B cells as immune complexes, as previously described for noninfected B cells ([67]), may only be poorly modelled in this system. Because EBV-infected cells down-regulate CD21, the extent to which EBV-infected B cells bind immune-complexed HIV-1 via CD21 and thus facilitate infection of susceptible cells, as described previously for non–EBV-infected B cells will have to be further evaluated. Furthermore, this model system cannot fully recapitulate the spatioanatomic aspects of human viral infection and lymph nodes and mucosal secondary lymphoid tissues are poorly developed. In this study, we focused our investigations on particular recombinant strains of type 1 EBV (B95-8) and X4-tropic HIV-1 (NL4-3), respectively. It will be necessary to validate these findings using other EBV and HIV-1 strains in future studies to reflect the strain diversity of both viruses in infected human individuals. Interestingly, type 2 EBV has been described to infect T cells in vitro, in humanized mice, and in healthy infected children, thus expanding further the cellular repertoire within which direct interaction of these two important human pathogens could occur ([68], [69], [70]).

# Materials and Methods

### Humanized mouse generation and infection

NOD/LtSz-scid IL2Rγnull-tgA2 (NSG-A2, Stock# 009617) and NOD/LtSz-scid IL2Rγnull (NSG, Stock# 005557) mice were obtained from The Jackson Laboratory, bred, and maintained at the Institute of Experimental Immunology, University of Zurich, under specific pathogen-free conditions, with a maximum of five adult animals per cage. Newborn mice (1–5 d old) were irradiated with 1 Gy. 5–7 h after irradiation, mice were injected with 1–3 × 10^5 CD34[+] human HPCs derived from HFL tissue obtained from Advanced Bioscience Resources. Isolation of human CD34[+] cells from HFL tissue was performed as described previously ([27], [71]) by positive selection for CD34 with magnetic separation according to the manufacturer's recommendations (Miltenyi Biotec). Both male (n = 48) and female (n = 80) mice were used. Individual mouse cohorts were reconstituted with cells derived from six different donors and NSG-A2 mice were reconstituted specifically with CD34[+] cells from HLA-A2[+] HFL samples. Reconstitution of human immune system components in mice was analyzed 3 mo after HPC injection and again 1 wk before the start of the experiments by flow cytometric immune phenotyping of PBMCs for huCD45, huCD3, huCD19, huCD4, huCD8, and HLA-DR as previously described ([72]). Apart from weighing and tail vein bleeding for analysis of human immune cell reconstitution in peripheral blood, humanized mice were not involved in any procedures before viral infection. Once human immune cell reconstitution was confirmed, humanized mice were injected with 10^4 Raji Infecting Units (RIUs) of EBV or PBS, and 1 wk later with 0.25 × 10^6 tissue culture infecting dose 50 (TCID$_{50}$) of HIV-1 or PBS intraperitoneally. For CD8[+] T-cell depletion, mice received intraperitoneal injections of purified anti-CD8 antibody, clone OKT-8.SUP obtained from hybridoma culture in-house ([73]) diluted in PBS (total of 150 μg per mouse) at week 2 (1 wk after HIV infection). Experiments were performed with a cohort of mice reconstituted with CD34[+] cells from a single donor, and animals were distributed to experimental groups to ensure a similar ratio of females to males (62.5% females to 37.5% males) and overall similar human immune reconstitution in the peripheral blood. Animals were euthanized 4 wk post-EBV infection. Investigators were not blinded regarding the viruses used for infection of the animals. Virus-inoculated mice with no EBV DNA load in the blood and spleen and without nuclear EBNA2 staining in formalin-fixed, paraffin-embedded (FFPE) splenic sections or with

splenic lymphocytes were MACS separated into CD19[+] and CD19-depleted fractions. Recipient mice received either CD19[+] or CD19-depleted cells i.p. from either EBV/HIV or EBV/HIV OKT8-treated mice. Recipient mice were monitored up to week 12 for HIV-1 RNA in plasma and for EBV DNA in whole blood. **(E)** The highest number of HIV-1 RNA copies measured in plasma of individual-recipient mice and the frequency of recipients with detectable HIV RNA in the plasma and EBV DNA in the peripheral blood after receiving either CD19[+] or CD19-depleted cells from either EBV/HIV or EBV/HIV OKT8-treated donor mice.

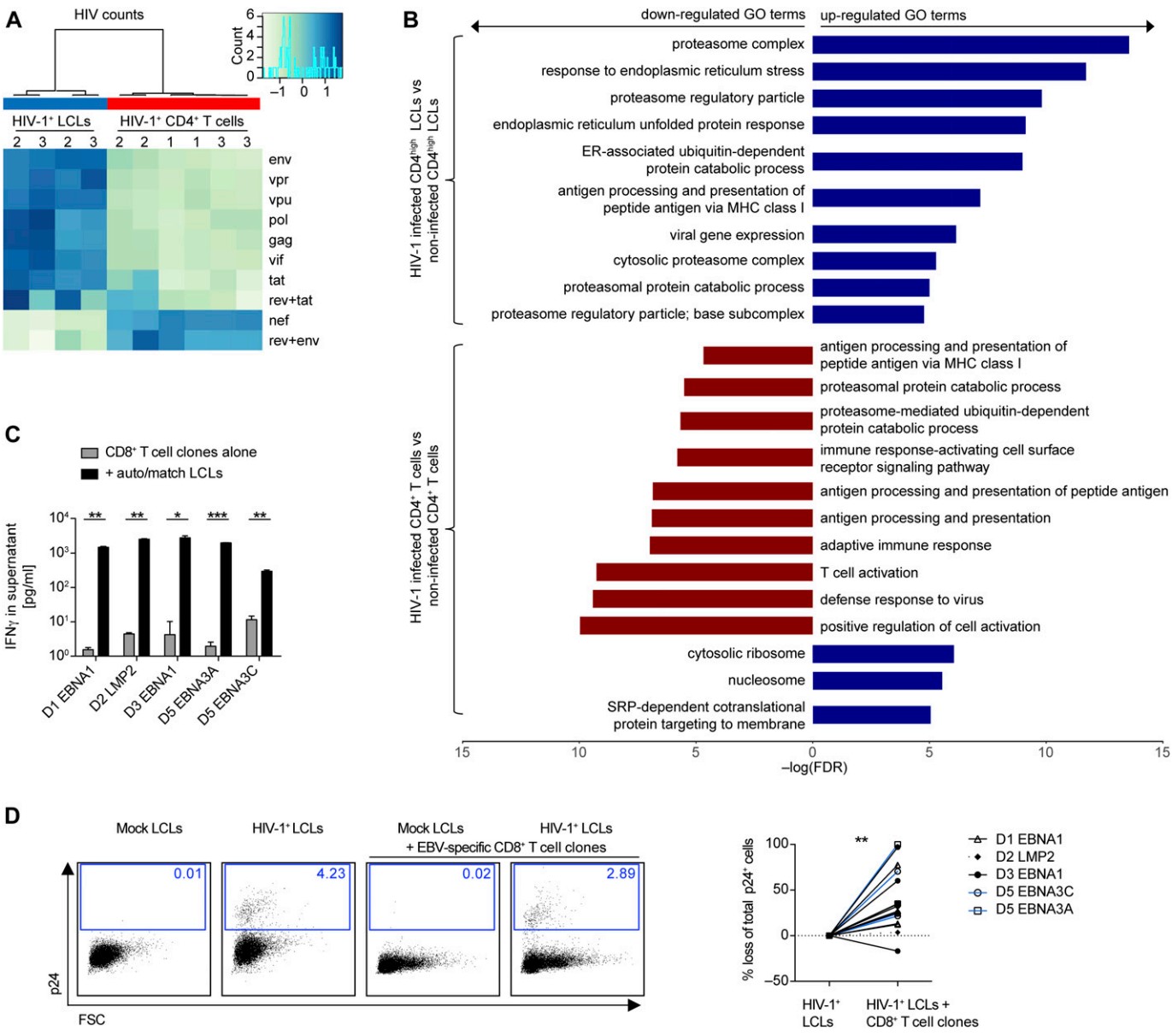

**Figure 5. Susceptibility of HIV-1–infected lymphoblastoid cell lines (LCLs) to T-cell clones in vitro.**
**(A)** Heat map depicting the HIV transcriptome in HIV-1–infected LCLs (2 donors) and three PB CD4⁺ T cells. Two of the CD4⁺ T cells investigated were autologous to the LCLs. Two separate infections were performed for each cell type. **(B)** Bar chart depicting selected results from gene set enrichment analysis of HIV-1–infected T cells and LCLs relative to mock-infected controls. **(A, B)** RNA was extracted 2 d after HIV-1 infection. **(C)** Reactivity of 5 EBV-specific CD8⁺ T-cell clones from four donors with or without autologous or HLA-matched LCLs (auto/match) measured by IFNγ ELISA of the culture supernatant. Donor and specificity for EBV protein: D1 EBNA1, D2 LMP2, D3 EBNA1, D5 EBNA3A, and D5 EBNA3C. Mean ± SD (two-tailed unpaired t tests, corrected by the Holm–Sidak method). **(D)** Representative flow cytometry plots from intracellular staining for p24 of mock-infected LCLs, HIV-1–infected LCLs, mock-infected LCLs mixed with an autologous EBV-specific CD8⁺ T-cell clone and HIV-1–infected LCLs mixed with an autologous EBV-specific CD8⁺ T-cell clone indicating percentage of p24-positive of total LCL population. Plots were pre-gated on live, single, total LCLs-labelled before co-culture with a lipophilic membrane dye (PKH67). Specific elimination of p24⁺ autologous or HLA-matched LCLs by individual EBV-specific CD8⁺ T-cell clones expressed as percentage loss of % p24⁺ cells in cocultures with T-cell clones compared with conditions without. Pooled from eight individual experiments, each condition was performed with 2–4 technical replicates (Wilcoxon matched pairs test). **(C, D)** P-values are reported as ns P > 0.05, *P < 0.05, **P < 0.01, ***P < 0.001.

no HIV RNA load in the peripheral blood and HIV cytoplasmic p24 staining in FFPE splenic sections at 4 wk postinfection were considered noninfected. The tumor frequency in humanized mice was assessed by a tumor score: no macroscopic tumors observed = 0; tumor observed = 1; multiple tumors observed = 2. For autologous cell transfer into humanized mice, see the following paragraphs.

## HIV and EBV virus production

HIV viral stocks were produced by polyethylenimine-mediated transfection (Polysciences) of 293T cells with X4-tropic NL4-3 and HBX2, R5-tropic YU-2 and JR-CSF, and dual-tropic 89.6 plasmid DNA, respectively, provided through the NIH AIDS Research and Reference

Reagent Program. 48 h after transfection, virus was harvested, filtered (0.45 µm, TPP syringe filter) and frozen at −80°C until use. Virus titers were determined as previously described (74). Briefly, $TCID_{50}$ was determined by infecting human $CD8^+$ T-cell–depleted PBMCs from three donors, which were stimulated by PHA and anti-CD3 beads (Dynal 11131D; Life Technologies). p24 in the postinfection culture supernatants were determined via ELISA. EBV B95-8-GFP (EBVwt, p2089 (75)) was produced in 293 HEK cells and titers of virus concentrates (in RIU) were determined by flow cytometric analysis of $GFP^+$ Raji cells 2 d after infection in vitro on a BD FACSCanto II (BD Biosciences) as previously described (72, 76).

## Primary cell cultures

PBMCs were obtained from whole blood of donors after red blood cell removal by density-gradient centrifugation using Ficoll-Paque (GE Healthcare) following the manufacturer's instructions. $CD19^+$ B cells were isolated from PBMCs by magnetic selection using the MACS $CD19^+$ isolation kit according to the manufacturer's instructions (Miltenyi Biotec) after prior incubation with the FcR blocking reagent (included in the $CD34^+$ Isolation Kit; Miltenyi Biotec) for 15 min at 4°C. Two MACS columns were used in succession. For the HIV-1 integration site and RNA transcriptome analysis, donor $CD4^+$ T cells were isolated by negative magnetic selection using the EasySep human $CD4^+$ T-cell isolation kit (Cat. no. 17952; Stemcell Technologies) following the manufacturer's instructions. LCLs and CD19-depleted PBMCs were cultured in RPMI 1640 (Gibco) medium supplemented with 10% FCS (PAA Laboratories), 50 U/ml penicillin–streptomycin, 25 mM Hepes, 2 mM L-glutamine (R10), and in the case of PBMCs and T cells, 20 IU/ml recombinant IL-2 (Peprotech) was included. $CD19^+$ B cells were cultured as previously described (77) in RPMI 1640 (Gibco) medium supplemented with 10% FCS (PAA Laboratories), 50 U/ml penicillin–streptomycin, 25 mM Hepes, 2 mM L-glutamine, IL-4 (10 ng/ml), IL-2 (50 ng/ml), IL-10 (10 ng/ml), and sCD40L (300 ng/ml). All cells were cultured in a humidified incubator at 37°C and 5% $CO_2$.

## HIV infection of cells in vitro

For in vitro HIV infections of CD19-depleted PBMCs, purified $CD4^+$ T cells were stimulated overnight in R10 supplemented with 5 µg/ml PHA and 20 IU recombinant human IL-2 (Peprotech). Cells were resuspended in HIV virus stocks pre-diluted in R10 at a multiplicity of infection of 1. Cells were centrifuged with the virus at 1,200$g$ for 2 h at 24°C, and then washed in PBS. CD19-depleted PBMCs and purified $CD4^+$ T cells were maintained in R10 supplemented with 20 IU/ml of recombinant IL-2 (Peprotech). LCLs were maintained in R10. $CD19^+$ B cells were maintained in R10 supplemented with IL-4 (10 ng/ml), IL-2 (50 ng/ml), IL-10 (10 ng/ml), and sCD40L (300 ng/ml) as described previously (77). HIV p24 antigen levels in cell culture supernatants were determined using a twin-site sandwich enzyme-linked immunosorbent assay (ELISA) performed essentially as described previously (78). Briefly, a polyclonal antibody was adsorbed to a solid phase to capture p24 antigen from detergent lysates of culture supernatants. Bound p24 was visualized with an alkaline phosphatase–conjugated anti-p24 monoclonal antibody and luminescent detection system. Time point zero for the p24

assay was collected after the final washing step. For in vitro ART experiments, the medium was supplemented with 1 µM Efavirenz and 5 µM AZT (provided through the NIH AIDS Research and Reference Reagent Program) from day 5 postinfection until the end of the experiment on day 10.

## CCR5 genotyping of LCLs

Genotyping was performed as previously described by reference 79. Briefly, PCRs were performed using 1 µl of donor LCL derived DNA and primers flanking the 32-bp deletion: 5'-GTCTTCATTACACCTGCAGCTCTC-3' and 5'-GTCCAACCTGTTAGAGCTACTGC-3' (79). DNA was extracted from a known CCR5Δ32 heterozygous individual and included as a control. The amplified products (wild-type CCR5: 311 bp and CCR5Δ32: 279 bp) were analyzed on a 3.0% agarose/TAE gel by electrophoresis and visualized with gel red DNA staining.

## HIV-1 integration site analysis

Genomic DNA and total RNA were extracted from HIV-1 NL4-3–infected and noninfected $CD4^+$ T cells and $CD4^{high}$ LCLs (60% $CD4^+$) with DNeasy Blood & Tissue kit and RNeasy Mini kit (QIAGEN), respectively, according to the manufacturer's protocol 2 d post-HIV-1 infection. 5' HIV-1 integration junctions were amplified from 400 to 1,300 ng of genomic DNA with the nrLAM-PCR (Table S6) (36, 37). A total of 12 pM cell type–specific pooled amplicons were sequenced with the Illumina MiSeq platform using the MiSeq Reagent Kit v2 (300 cycles) (Illumina) with 8% PhiX. Sequencing reads were mapped to the human genome assembly GRCh37.p13 using the Integration Site Analysis Pipeline (InStAP) (37). True 5' HIV-1 integration junctions were identified based on the following criteria: (i) presence of nrLAM-PCR adaptor and 5' HIV-1 LTR in the sequencing reads, (ii) host DNA begins within three nucleotides from the end of 5' HIV-1 LTR, and (iii) ≥98% identity from ≥85% of the sequencing read lengths. Testing for integration of HIV into the EBV DNA genome was performed by utilizing the same bioinformatics pipeline InStAP, exchanging the target genome from the human genome, to the EBV genome (National Center of Biotechnology Information [NCBI] taxid:10376). The default settings for BLAST did not return any hits. The use of BLASTn (suitable for similar sequences) resulted in some hits with similarities that were, however, not statistically significant. Nucleotide conservation plots were prepared using WebLogo (80, 81). Overrepresentation of GO terms among HIV-1 integration sites in genes in NL4-3–infected LCLs and $CD4^+$ T cells was performed with a PANTHER Overrepresentation Test (Version 20171205) with the GO database version (2018-02-02) with Fisher's exact test with FDR multiple test correction.

## RNA transcriptome analysis

RNA was extracted from cells as described in "HIV-1 integration site analysis." High-throughput sequencing and RNA-seq analysis was performed essentially as previously described (49). Sequenced reads were mapped to a merged genome consisting of EBV (NC_007605.1), HIV-1 (AF324493.2), and Homo Sapiens (GRCh38.p7) using STAR (v2.5.2a) (82). The option "--quantMode GeneCounts" was enabled to count the reads per gene simultaneously, resting

upon the annotations from GenBank (EBV and HIV-1) and Ensembl Release 86 (Homo Sapiens). Based on these counts, statistical analysis of differential gene expression was performed with R/Bioconductor package DESeq2 (v1.16.1) (83) because low HIV-1 infection LCL-sample 3–11 was excluded. The obtained P-values were corrected for multiple testing by the Benjamini–Hochberg procedure, genes with a mean absolute fold change of minimal 1.5 and a corrected P-value lower than 0.1 were considered differentially expressed. For gene set enrichment analysis, R/Bioconductor package gage (v2.20.1) was applied (84), gene sets were obtained from the Gene Ontology Consortium. Gene sets with a Benjamini–Hochberg–corrected P-value lower than 0.001 were considered enriched. Heat maps were generated with the normalized counts provided by DESeq2 using R package gplots (v3.0.1). Non–HIV-1–dependent differences between the donors have been removed with R/Bioconductor Package sva (v3.22.0) before plotting, using HIV-1 infection as a surrogate variable. The complete RNA-seq data from this article have been deposited in the European Nucleotide Archive (http://www.ebi.ac.uk/ena). The accession number for the RNA-seq data is PRJEB25772.

## Quantification EBV and HIV burden in humanized mice

EBV infection in mice was monitored by qPCR of DNA from splenic tissue and whole blood. DNA was extracted using DNeasy Blood and Tissue Kit (QIAGEN) and NucliSENS (BioMerieux), respectively, according to the manufacturer's instructions. Quantitative analysis of EBV DNA in humanized mouse spleens and blood was performed by a TaqMan (Applied Biosystems) real-time PCR as described previously (85) with modified primers (5′-CTTCTCAGTC-CAGCGCGTTT-3′ and 5′-CAGTGGTCCCCCTCCCTAGA-3′) and the fluorogenic probe (5′-(FAM)-CGTAAGCCAGACAGCCAATTGTCAG-(TAMRA)-3′) for the amplification of a 70-base pair sequence in the conserved BamHI W fragment of EBV. The PCR was run on an ABI 7300 Thermocycler (Applied Biosystems) and samples analyzed in duplicates. HIV infection in humanized mice was assessed by RT-qPCR for HIV copy numbers in plasma or serum (AmpliPrep/COBAS TaqMan HIV-1 test; Roche). HIV-1 copy numbers in plasma of non-reconstituted NSG mice transplanted with HIV-infected LCLs and humanized controls were assessed by an in-house RT-qPCR assay. 60 $\mu$l of plasma or serum of mice was diluted in 80 $\mu$l of water. RNA was extracted following the QIAamp Viral RNA Mini Kit manufacturer's instructions (QIAGEN). Reverse transcription was performed using 12 $\mu$l of RNA using the SKCC1B primer 5′-TACTAGTAGTTCCTGCTATGTCACTTCC-3′ (86). The reaction was performed in a final volume of 20 $\mu$l:0.25 $\mu$M SKCC1B primer, 4 $\mu$l 5× iScript selection reaction mix, 2 $\mu$l gene specific primer enhancer solution, 1 $\mu$l iScript reverse transcriptase (Bio-Rad). The thermal cycler was set to 42°C for 60 min followed by inactivation at 85°C for 5 min. The real-time quantitative PCR was performed using 3 $\mu$l of cDNA prepared during the reverse transcription step as previously described (87) using the ts5′gag 5′-CAAGCAGCCATGCAAATGT-TAAAAGA-3′ and SKCC1B primers as well as the mf319tq 5′-TGCAGCTTCCTCATTGATGGT-3′ probe. The real-time quantitative PCR reaction was performed in a final volume of 12.5 $\mu$l:3.8 $\mu$M of each primer, 1.15 $\mu$M of the probe, and 6.25 $\mu$l DreamTaq Hot Start PCR master mix. Thermal cycling was performed with a C1000 Touch

CFX384 Real-Time platform (Bio-Rad) starting with 95°C for 4 min and 50 cycles of the following steps: denaturation at 95°C for 5 s, annealing at 55°C for 5 s, and elongation at 60°C for 30 s.

## Serum cytokine quantification

Serum aliquots were collected during euthanizing animals and stored at –80°C until use. Cytokine concentrations in the serum were determined with the V-plex Mesoscale Discovery Human Proinflammatory Panel 1 (Cat. no. K15049D) and was performed as per the manufacturer's instructions in duplicates; standard dilutions of the calibrator blend were performed in quadruplicates.

## Flow cytometry

Total white blood cells in whole blood were counted using a Beckman cell counter according to the manufacturer's instructions. Mouse whole blood was collected in K2 EDTA tubes (BD) and lysed twice with ACK lysis buffer (Lonza) for 5 min and washed in PBS. Mouse splenocyte cell suspensions were obtained by mechanical disruption of the spleen through a 70-$\mu$m nylon mesh, lysis with ACK lysis buffer for 10 min, and washing with PBS. Cells from mouse whole blood and spleens or in vitro–cultured cells were labelled with mixtures of fluorochrome-labelled antihuman monoclonal antibodies in PBS (Table S7). Acquisitions were performed on an LSR Fortessa (BD) flow cytometer and data were analyzed with FlowJo (v9). Cellular debris and dead cells were excluded by their light-scattering characteristics, as well as fixable live dead stains.

## RT-qPCR

Total RNA was isolated from in vitro–infected B cell lines or humanized mice derived CD19$^+$ splenocytes using the RNeasy Mini Kit (QIAGEN) according to the manufacturer's instructions. Contaminating genomic DNA was removed with a 15-min on-column DNase treatment during RNA isolation (RNase-Free DNase Set; QIAGEN). The purified RNA was immediately reverse-transcribed with GoScript Reverse Transcriptase (Promega) according to the manufacturer's recommendations in a 20-$\mu$l volume for 1 h at 42°C using a primer mix combining previously described 3′ gene-specific RT primers (53, 88) and reverse primers for HIV transcripts listed in Table S6 at concentrations of 10 $\mu$M each. After 15-min heat inactivation at 70°C, 0.2 U RNase H (Thermo Fisher Scientific) was added for 20 min at 37°C, followed by RNase H heat inactivation for 20 min at 65°C and freezing of the cDNA at –20°C. Amplifications of Cp/Wp-EBNA1, EBNA2, LMP1, LMP2A, BZLF1, GAPDH, and SDHA were carried out in triplicate with equal volumes of input RNA and TaqMan universal PCR reagents (Applied Biosystems) using either previously published primer sets with 5′FAM/3′TAMRA (88) or 5′FAM/3′MGB labelled probes for SDHA (TaqMan Applied Biosystems Gene Expression Assay [Hs00417200]). Primers and probes for quantification of HIV-unspliced, unspliced and single-spliced, and multiple-spliced transcripts (89, 90) are listed in Table S6. Thermal cycling was performed with a C1000 Touch CFX384 Real-Time platform (Bio-Rad) starting with 2 min at 50°C and 10 min at 95°C, followed by 50 cycles of amplification (95°C for 15 s and 60°C for 1 min). Cq values were determined with the CFX-manager

software (Bio-Rad) using a regression algorithm. Levels of EBV transcripts in B cells from infected humanized mice were calculated relative to the geometric mean of the two reference genes, *GAPDH* and *SDHA*, and then normalized to the mean value of the EBV[+] single-infected animals. Mock-infected control animals were negative for EBV transcripts. For HIV-1 splice variant RNA quantification in infected LCLs, absolute HIV-1 transcript numbers were calculated using dilutions of an all-in-one HIV-1 standard DNA containing the HIV-1 RNA splice variants as described and calculated previously ([91]).

### EBV-specific IFN-γ release assay (ELISPOT)

EBV-specific T-cell responses were analyzed using an IFNγ ELISpot assay, essentially as previously described ([27], [92]). Briefly, splenocytes were depleted of human CD19[+] cells using anti-CD19 microbeads (Miltenyi Biotec). The CD19-depleted fraction was stimulated with autologous LCLs at a ratio of 1:4 for 18 h and incubation with R10 and PMA/ionomycin served as negative and positive controls, respectively. Each condition was performed in duplicates or triplicates. Spots were counted on an ELISPOT reader system (ELR02; Autoimmun Diagnostika GmbH).

### Immunohistochemistry

Immunohistochemistry (IHC) was performed on sections obtained from FFPE-humanized mouse spleen and tumor tissue with a BOND-MAX automated IHC system (Leica Microsystems). Sections of 3 μm from FFPE blocks were deparaffinized, and then antigen retrieval was performed by incubation in Bond Epitope Retrieval Solution 2 (ER2; Leica) for 30 min at 100°C followed by incubation with antibodies against p24 (M0857, dil. 1:40; Dako), EBNA2 (Ab90543, dil. 1:200; Abcam Ltd.), CD20 (120R-16, dil.1:300; Cell Marque Lifescreen Ltd.), CD4 (104R-16, dil.1:100; Cell Marque Lifescreen Ltd.), or CD8 (108R-16, 1:500; Cell Marque Lifescreen Ltd.), respectively, which were detected with the Refine-DAB-Kit (Leica). For double IHC of p24 and EBNA2, DAB[+] p24–stained sections were incubated in ER2 for 10 min at 95°C before incubation with the mouse anti-EBNA2 antibody, which was detected with the Refine AP-Kit and new fast red (Leica). For double IHC of PAX5 and p24, sections were incubated in ER2 for 20 min at 95°C before incubation with the mouse anti-PAX5 antibody (PA0552, dil. 1:20; Leica) for 30 min, which was detected with the Bond Polymer Refine Detection Kit and DAB (Leica), then incubated with the mouse anti-p24 antibody (M0857, dil. 1:10; Dako) for 30 min, which was detected with the Bond Polymer Refine Red Detection Kit and new fast red (Leica). Nuclei were counterstained with hematoxylin (AP refine Kit; Leica or JT Baker Hämatoxylin; Leica). EBER in situ hybridization and CD4 co-immunohistochemistry was performed on 3 μm FFPE PTLD sections from a tissue microarray block ([34]) using a Benchmark Ultra automated slide stainer (Ventana). Briefly, tissue sections were pretreated with protease 3 before incubation with an EBER-specific probe (Ventana), which was detected via the ISH iView Blue Detection Kit (Ventana). Subsequently, tissue sections were incubated with cell conditioner 1 for antigen retrieval, rabbit monoclonal antibody against human CD4 (clone SP35), and visualization performed with OptiView DAB

IHC Detection Kit (Ventana). All images presented in this article were white set point adjusted with Adobe Ps CS5 V12.1.

### Quantification of immunohistochemistry

PAX5/p24 and EBNA2/p24 co-stainings were analyzed with a Vectra3 automated quantitative pathology imaging system (PerkinElmer) using Vectra and InForm software (PerkinElmer). Images were acquired via an automated scanning protocol created with InForm tissue segmentation to recognize the tissue. Images were acquired with 20× objective lens with a CCD camera using the scanning protocol. Images taken were used to set up algorithms in inForm to recognize and count PAX5/p24– or EBNA2/p24–positive cells, respectively. For comparison of tumor tissue and non-tumorous spleen, the regions of interest were defined manually. Single chromogen control stains were used to eliminate signal cross-talk. The number of positive cells was determined per 1 mm$^2$.

### Autologous transfer experiments

Once human immune cell reconstitution was confirmed, 3 mo after reconstitution, donor humanized mice were injected i.p. with 10$^5$ EBV RIU or PBS and 1 wk later with 5 × 10$^5$ NL4-3 HIV-1 TCID$_{50}$ or PBS. Mice received 150 μg OKT8 antibody (purification described below) or PBS i.p. at week 2 (1 wk after HIV infection) and at week 4. Animals were euthanized between 5 and 6 wk post-EBV infection. Spleens of donor mice were mashed through a 0.75 μM nylon mesh followed by red blood cell lysis using ACK buffer (Lonza) as per the manufacturer's instructions. CD19[+] B cells were isolated from total splenocytes as described for PBMCs above. The labelled B-cell fraction was passed over two MACS columns in succession. The non-labelled T-cell–containing CD19-depleted fraction was also collected. Purity was confirmed via flow cytometry immunophenotyping. Equivalent numbers of CD19[+] B cells or CD19-depleted splenocytes were injected into autologously reconstituted (same HFL donor-derived CD34[+] HPCs) recipient humanized mice (in one case the respective cell fractions from two mice were pooled). Autologous HPC-reconstituted recipient animals were chosen to avoid an allogenic T-cell response. Recipient mice were euthanized 6–7 wk post-donor cell transfer.

### HIV-1 latency experiments

Non-reconstituted 9–18-wk-old NSG mice received 1–2 × 10$^5$ in vitro NL4-3 HIV-1–infected LCLs via i.p. injection. LCLs were derived from EBV-infected humanized mice via ex vivo outgrowth in R10 medium. The B cell identity of these LCLs was confirmed via RNA-seq (Sample E4-3 ([53])). Control humanized mice or non-reconstituted NSG mice were injected i.p. with 5 × 10$^5$ NL4-3 HIV-1 TCID$_{50}$. cART was applied via the drinking water in MediDrop Sucralose solution, and briefly abacavir (ABC) (100 mg/kg/day), lamivudine (3TC) (150 mg/kg/day), and dolutegravir (DTG) (15 mg/kg/day) pills were reduced to powder and solubilized in MediDrop Sucralose solution for 15 min on a magnetic stirrer. Final amount of drugs per 500 ml of MediDrop Sucralose solution was 0.54 g of ABC, 0.622 g of 3TC, and 1.86 g of DTG. Control humanized mice received cART 4 wk post-HIV-1 infection for 2 wk and then were switched back to normal drinking

water for 2 wk. LCL-bearing mice received cART on the day of HIV-1⁺ LCL transfer for 2 wk and then were switched back to normal drinking water for 2 wk. All animals were euthanized 4 wk post-start of cART treatment. Spleens and tumor material of mice were disrupted mechanically through a 0.75-$\mu$M nylon mesh followed by red blood cell lysis of splenocytes using ACK buffer (Lonza) as per the manufacturer's instructions. Total genomic DNA (for ALU-gag LTR qPCR) and RNA (for HIV-1 copy number RT-qPCR) was simultaneously extracted using the RNeasy Midi Kit (QIAGEN) as previously described (93). Briefly, the disrupted spleen and tumor material was resuspended in RLT lysis buffer (QIAGEN) as per the manufacturer's instructions. After RNA isolation, genomic DNA was eluted from the RNeasy gel-silica membrane by addition of DNA- and nuclease-free isolation buffer (8.0 mM NaOH). Columns were left at 55°C for 10 min and centrifuged at 5,000$g$ for 3 min at room temperature. pH of samples was adjusted with Hepes to pH 7–8 and supplemented with 1.0 mM EDTA.

### HIV-1 Alu-PCR integration assay

Genomic DNA were extracted from splenocytes of HIV-1 NL4-3–infected humanized NSG mice and from tumors derived from NSG mice engrafted with HIV-1 NL4-3–infected LCLs. A two-step PCR amplification was performed as described (94). The first PCR amplification was performed on 300 ng of genomic DNA, or no template controls with the genomic *Alu* forward 5'-GCC TCC CAA AGT GCT GGG ATT ACA G-3' and HIV gag reverse 5'-GTT CCT GCT ATG TCA CTT CC-3' primers. The reactions were performed in 50 $\mu$l: 1× PCR buffer, 1.5 mM MgCl$_2$, 0.3 mM dNTPs, 0.1 $\mu$M *Alu* forward primer, 0.6 $\mu$M gag reverse primer, and 0.05 units of Jump start Taq DNA polymerase/$\mu$l (Sigma-Aldrich). The thermal cycler ABI 7500 (Applied Biosystems) was programmed to perform a 2-min hot start at 95°C and then 20 cycles of the following steps: denaturation at 95°C for 30 s, annealing at 50°C for 30 s, and elongation at 72°C for 3 min and 30 s with a final extension at 75°C for 5 min. The second-round real-time quantitative PCR was performed using 10 $\mu$l of the reaction from the first PCR amplification with the LTR (R) forward 5'-TTA AGC CTC AAT AAA GCT TGC C-3'; LTR (U5) reverse, 5'-GTT CGG GCG CCA CTG CTA GA-3' primers. The wild-type probe primer sequence was 5'-FAM-CCA GAG TCA CAC AAC AGA GGG GCA CA-TAMRA-3'. Reaction was carried out in 20 $\mu$l reaction mix that contained 1× PCR buffer, 3.5 mM MgCl$_2$, 0.005 $\mu$M carboxy-X-rhodamine (ROX; Molecular Probes), 0.3 mM dNTPs, 0.25 $\mu$M LTR forward and reverse primers, 0.1 $\mu$M wild-type probe, and 0.125 units of JumpStart Taq DNA polymerase/$\mu$l (Sigma-Aldrich). The reactions were performed on an ABI 7500 instrument (Applied Biosystems) with the following thermal program: 5 min hot start at 95°C and then 50 cycles of the following steps: denaturation at 95°C for 15 s and annealing and elongation at 60°C for 30 s. The ABI 7500 measurement point was at elongation 60°C-30 s.

### Purification of mouse antihuman CD8 antibody

The OKT-8 hybridoma cells (mouse IgG2a; ATCC) (73) were gradually adapted to the serum- and protein-free PFHM-II medium (Cat. no. 12040-077; Thermo Fisher Scientific, [formerly Gibco]). Adapted cells were cultured in BD CELLineTM 1000 flasks (Cat. no. 353137; BD) according to the manufacturer's recommendation. Antibodies were precipitated from the culture supernatant by addition of an equal

volume of saturated ammonium-sulfate solution. After buffer exchange to PBS using PD-10 desalting columns (Sephadex G25 medium; GE Healthcare), antibodies were sterilized by filtration through 0.2 $\mu$m (Filtropur S 0.2; Sarstedt) and stored at 4°C until use. Antibody purity was assessed by PAGE and Coomassie Brilliant Blue staining, protein concentration was determined by OD280 nm measurement (using an extinction coefficient of 1.37 (at 1 g/l) for concentration calculation) and CD8-binding was confirmed by flow cytometry.

### CD8⁺ T-cell cloning and restimulation

The LMP2$_{426–434}$–specific CD8⁺ T-cell clone that detects the CLGGLLTMV epitope presented on HLA-A*0201 used in this study has previously been described (72). CD8⁺ T-cell clones specific for EBNA1 were generated from freshly isolated PBMCs which were depleted of CD4⁺ T cells using anti-CD4 conjugated MACS microbeads and magnetic selection (Miltenyi Biotec). Additional CD8⁺ T-cell clones specific for EBNA3A and –C were generated without prior CD4⁺ T-cell depletion. PBMCs were stimulated with 5 $\mu$M of EBNA1 HPVGEADYFEY peptide or EBNA3A–C peptide mix (EBNA3A: FLRGRAYGL, RPPIFIRRL, YPLHEQHGM, RLRAEAQVK; EBNA3B: IVTDFSVIK; and EBNA3C: RRIY-DLIEL, EENLLDFVRF) for three to 4 h. Responding cells were enriched by IFN-γ secretion assay (130-054-201; Miltenyi Biotec) and were cloned by limiting dilution at 3 and 30 cells per well on HLA-matched gamma-irradiated LCLs (10⁴/well) loaded with the relevant peptide or peptide mix at 5 $\mu$M and allogeneic gamma-irradiated, PHA-treated PBMCs (10⁵/well) in IL-2-supplemented (150 IU/ml) medium with 10% pooled human sera (T cell medium). Growing microcultures were screened for peptide reactivity by IFN-γ ELISA (3420-1H-20; Mabtech). Selected CD8⁺ T-cell clones were expanded on autologous gamma-irradiated LCLs loaded with the relevant peptide at 5 $\mu$M and allogeneic gamma-irradiated, PHA-treated PBMCs in T-cell medium. Clones were maintained in the T-cell medium. Every 2–3 wk, CD8⁺ T-cell clones were restimulated with gamma-irradiated peptide-pulsed autologous LCLs and allogenic PHA-treated PBMCs. To control for specific recognition of target cells, 5,000 CD8⁺ T-cell clones were co-cultured with 50,000 autologous LCLs. After overnight incubation, co-culture supernatants from single wells were tested by ELISA for IFN-γ content. The resulting additional CD8⁺ T cell clones used in this study were specific for EBNA1$_{407–417}$ (HPVGEADYFEY, HLA-B*3501, clones from two donors), EBNA3A$_{325–333}$ (FLRGRAYGL, HLA-B*0850), and EBNA3C$_{281–290}$ (EENLLDFVRF, HLA-B*4402). The CD8⁺ T-clone–specific for Influenza MP$_{(58–66)}$ GILGFVFTL presented on HLA-A*0201 described previously (95) was kindly provided by Jean-Francois Fonteneau, Nantes, France. The CD8⁺ T clones D11 and D12 specific for HIV-1 gag peptide SLYNTVATL and HIV-1 pol peptide ILKEPVHGV presented on HLA-A*0201, respectively, were kindly provided by Yanchun Peng and Tao Dong, Oxford, UK. The CD8⁺ T clone D13 specific for HIV-1 gag peptide SLYNTVATL presented on HLA-A*0201 was kindly provided by Arnaud Moris, Paris, France.

### In vitro killing assay

Cytotoxic activity of CD8⁺ EBV-specific T-cell clones against autologous or HLA class I–matched HIV-1–infected LCLs was assessed by in vitro killing assays. X4-tropic HIV-1–infected LCLs and noninfected LCLs were labelled with a lipophilic membrane dye (PKH67,

Cat. no. MINI67; Sigma-Aldrich, or CellVue Claret Far Red, Cat. no. MINCLARET-1KT; Sigma-Aldrich) according to the manufacturer's instructions. Infected and noninfected LCLs were incubated with EBNA1-, EBNA3A-, EBNA3C-, or LMP2-specific CD8$^+$ T-cell clones for 18 h at an effector to the target ratio of 5:1. Cells were stained with the Zombie NIR live cell/dead cell discrimination dye (Cat. no. 423105; BioLegend) and treated with the BD fixation/permeabilization kit for intracellular staining of HIV-1 p24 core antigen using the HIV-1 p24 (Beckman Coulter) antibody. Specific lysis of HIV$^+$ LCLs was determined by using the formula: % lysis = 100 ×([% p24$^+$ of PKH67$^+$ LCLs without effectors − % p24$^+$ of PKH67$^+$ LCLs with effectors]/% p24$^+$ PKH67$^+$ LCLs without effectors). All culture conditions were run at least in duplicates and when possible in triplicates, or higher replicates.

## Statistics

Data were analyzed using GraphPad Prism software (v8.3.1). Normally distributed data were analyzed with a two-tailed $t$ test, with Welch's correction if variances were significantly different as determined by the f test. Paired data sets were compared with a two-tailed paired $t$ test or Wilcoxon matched pairs test. Mann–Whitney test was used for non-normally distributed data to compare the means of data sets. Multiple $t$ tests were corrected by the Holm–Sidak method. Two-tailed chi-squared (n > 10,000) test or Fisher's exact test (n < 10,000) was used for categorical data. Differences were considered significant at $P < 0.05$ in two-tailed tests. For analysis of RNA sequencing data, see RNA transcriptome analysis.

## Study approval

Studies using HFL samples as well as peripheral blood samples of healthy HIV-negative donors were reviewed and approved by the Cantonal Ethics Committee of Zurich, Switzerland (protocols no. KEK-StV-Nr.19/08 and 2019-00837). Studies using blood samples of EBV$^+$ PTLD donors and EBV$^+$ infectious mononucleosis or control donors, upon receipt of written informed consent, were reviewed and approved by the Institutional Review Board of the University of Hong Kong/Hospital Authority Hong Kong West Cluster (IRB reference no. UW 16-508 and EC 1940-02). The study of PTLD patient tissue microarray material was reviewed and approved by the Ethics Committee of Northwestern and Central Switzerland (protocol no. EKNZ 2014-252). Research was conducted in accordance with the Declaration of Helsinki. Animal protocols were approved by the veterinary office of the canton of Zurich, Switzerland (protocol nos. 148/2011, 209/2014, 93/2014, and 81/2017).

# Supplementary Information

# Acknowledgements

The authors thank Arbeneshe Berisha of Kempf und Pfaltz Histologische Diagnostik, Switzerland, for technical support with PAX5/p24 dual-IHC; Petra Hirschmann of the Institute of Pathology, University Hospital of Basel, Switzerland, and Silvia Behnke of Sophistolab AG, Switzerland, for providing additional IHC stainings; and current and past members of the Münz, Speck, and Metzner laboratories for technical support. This study was supported by Cancer Research Switzerland (KFS-3234-08-2013 and KFS-4091-02-2017); Worldwide Cancer Research (14-1033); SPARKS (15UOZ01), KFSP$^{MS}$, KFSP$^{HHLD}$, and KFSP-Precision$^{MS}$ of the University of Zurich; the Sobek Foundation; the Vontobel Foundation; the Baugarten Foundation; the Swiss Vaccine Research Institute; the Swiss MS Society; Roche; ReiThera; and the Swiss National Science Foundation (310030_162560, 310030B_182827, CRSII3_160708, and CRSII5_180323) to C Münz (310030_141067) and to KJ Metzner and the KFSP$^{HHLD}$ of the University of Zurich to A Aguzzi, MG Manz, RF Speck, and C Münz. D McHugh was supported by an MD-PhD fellowship from the Swiss Academy of Medical Sciences and Swiss National Science Foundation (323530_145247). R Myburgh and N Caduff were supported by the Forschungskredit of the University of Zurich.

## Author Contributions

D McHugh: conceptualization, formal analysis, funding acquisition, investigation, visualization, and writing—original draft, review, and editing.
R Myburgh: conceptualization, formal analysis, funding acquisition, investigation, visualization, and writing—original draft, review, and editing.
N Caduff: investigation.
M Spohn: formal analysis, investigation, and visualization.
YL Kok: formal analysis, investigation, and visualization.
CW Keller: investigation.
A Murer: formal analysis.
B Chatterjee: investigation.
J Rühl: investigation.
C Engelmann: investigation.
O Chijioke: investigation.
I Quast: investigation.
M Shilaih: formal analysis.
VP Strouvelle: resources.
K Neumann: investigation.
T Menter: resources.
S Dirnhofer: resources.
JKP Lam: resources.
KF Hui: resources.
S Bredl: resources.
E Schlaepfer: resources.
S Sorce: investigation.
A Zbinden: investigation.
R Capaul: investigation.
JD Lünemann: supervision.
A Aguzzi: supervision and funding acquisition.
AKS Chiang: resources.
W Kempf: resources and investigation.
A Trkola: conceptualization and writing—review and editing.
KJ Metzner: resources, supervision, and writing—review and editing.
MG Manz: resources, supervision, and funding acquisition.
A Grundhoff: formal analysis and supervision.
RF Speck: conceptualization, resources, supervision, funding acquisition, and writing—review and editing.
C Münz: conceptualization, resources, supervision, funding acquisition, and writing—original draft, review, and editing.

**Life Science Alliance**

## Conflict of Interest Statement

RF Speck and R Myburgh are shareholders of Transcure Bioservices (http://www.transcurebioservices.com/), a service provider using HIV-1–infected humanized mice for preclinical drug testing. The remaining authors have declared that no conflict of interest exists.

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
