## [Reviewer comments · Life Science Alliance]

Life Science Alliance

EBV renders B cells susceptible to HIV-1 in humanized mice

Donal McHugh, Renier Myburgh, Nicole Caduff, Michael Spohn, Yik Lim Kok, Christian Keller, Anita Murer, Bithi Chatterjee, Julia Rühl, Christine Engelmann, Obinna Chijioke, Isaak Quast, Mohaned Shilaih, Victoria Strouvelle, Kathrin Neumann, Thomas Menter, Stephan Dirnhofer, Janice Lam, Kwai Hui, Simon Bredl, Erika Schlaepfer, Silvia Sorce, Andrea Zbinden, Riccarda Capaul, Jan Lünemann, Adriano Aguzzi, Alan Chiang, Werner Kempf, Alexandra Trkola, Karin Metzner, Markus Manz, Adam Grundhoff, Roberto Speck, and Christian Munz

DOI: <https://doi.org/10.26508/lsa.202000640>

Corresponding author(s): Christian Munz, University of Zurich and Donal McHugh, University of Zurich

Review Timeline:

Submission Date:	2020-01-06
Editorial Decision:	2020-02-04
Revision Received:	2020-05-15
Editorial Decision:	2020-06-09
Revision Received:	2020-06-10
Accepted:	2020-06-11

Transaction Report:

February 4, 2020

Re: Life Science Alliance manuscript #LSA-2020-00640-T

Prof. Christian Munz
University of Zurich
Viral Immunobiology Institute of Experimental Immunology University of Zuerich
Winterthurerstrasse 190
Zuerich, Zurich CH-8057
Switzerland

Dear Christian,

Thank you for submitting your manuscript entitled "EBV renders B cells susceptible to HIV-1 in humanized mice" to Life Science Alliance. The manuscript was assessed by expert reviewers, whose comments are appended to this letter.

As you will see, the reviewers appreciate your analyses. However, they are not convinced by the proposed mechanism involving CD4 upregulation and think that the involvement of other receptors should get considered. Looking at correlations with CR2 expression could address this point. The reviewers also note some other inconsistencies that need addressing.

Based on the reviewer input and cross-comments received, we would like to invite you to submit a revised version of your manuscript to us. Importantly,

- the requested statistical analyses and quantifications should get performed (rev#1)
- alternative explanations should get considered (rev#1-3)
- comparisons to previous data sets available should get performed and the discrepancies need to get discussed / alternative mechanisms considered
- some drawbacks of the present work should get more openly discussed
- the conclusion on the impact of CD8+ T cells, particularly in the passive transfer of CD19+ T cells from the co-infected mice seems inconsistent and needs clarifying

The typical timeframe for revisions is three months. Please note that papers are generally considered through only one revision cycle, so strong support from the referees on the revised

version is needed for acceptance.

Thank you for this interesting contribution to Life Science Alliance. We are looking forward to receiving your revised manuscript.

Sincerely,

B. MANUSCRIPT ORGANIZATION AND FORMATTING:

*****IMPORTANT:** It is Life Science Alliance policy that if requested, original data images must be

made available. Failure to provide original images upon request will result in unavoidable delays in publication. Please ensure that you have access to all original microscopy and blot data images before submitting your revision.***

Reviewer #1 (Comments to the Authors (Required)):

In this manuscript, the authors investigated an unexpected change in cellular tropism of HIV1 in the context of Epstein Barr virus (EBV) co-infection. Notably, the authors showed that EBV-transformed B cells started to upregulate the HIV-coreceptors CD4 and CXCR4 and became susceptible to X4-tropic HIV infection in vitro. McHugh and colleagues subsequently confirmed a similar profile of proviral integrants in LCL and CD4 T cells. To confirm their observations in vivo, the authors took advantage of the only challenge model for HIV and EBV, namely human CD34+ HSC injected NSG mice which engrafted with components of a human immune system. As previously reported and consistent with clinical observations HIV infection reduced the numbers of CD4 helper T cells but not of CD8 T cells. McHugh et al provide evidence that CD8+ T cells expand in numbers upon co-infection but fail to control EBV in dually infected animals. Interestingly, transfer of B cells extracted from HIV/EBV co-infected humanized mice resulted in HIV infection in naïve animals. While there is currently no evidence from the clinical literature suggesting that B cells from EBV/HIV co-infected patients are actually sensitive to HIV infection clinically, the data presented here are certainly intriguing and provide a strong rationale for more mechanistic follow-up studies. Overall, the manuscript is well written, the experiments are well designed and the data largely support the claims by the authors.

Additional comments.

- Please apply stringent statistical analysis to all data presented to affirm significance.
- Do the authors have any data that would possibly hint a mechanism for the upregulation of CD4/CXCR4 in LCLs? Have the authors analyzed other (cytokine) receptor loci to ascertain how specific the change in transcription is to CD4/CXCR4?
- Do the transformed B cells acquire any other markers of T helper cells (phenotypic or functional)?
- Figure 1

A. Please provide the data of at least 2 donors

B. This reviewer is wondering why the kinetics of p24 differ between human donor-derived LCLs and NSG-HIS mice derived LCLs. This indicates their humanized mice does not mimic the actual human behavior. Please discuss and provide the data of two additional mice with RNA copy on 20, 25 days post infection.

- Fig.2

There is a considerable difference in the number of samples (CD4+ T cells vs LCLs) that were included in this analysis. Why are the numbers so heavily skewed towards LCLs? Also, it would have been desirable to compare the integration profile of LCL with CD4+high and CD4+low expressing cells.

- Fig.3

B. The reviewer is not convinced that CD8+ T cell frequencies are actually decreased during co-infection based on the histological data provided. Please provide quantitative data.

- Fig.5

o The data of "HIV-infected LCL vs non-infected LCLs" is not so helpful. LCLs should be stratified by expression levels of CD4+ since the phenotype of "bulk LCLs" is dissimilar to actual CD4+T cells.

o What was the rationale for not comparing HIV-infected LCL (CD4+high) and non-infected LCL (CD4+ high)?

Minor comments:

1. Page 13: Please add "in vitro" after the sentence "Furthermore, we found that dual-infected mice..."

Reviewer #2 (Comments to the Authors (Required)):

The manuscript by McHugh and Myburgh and colleagues describes an interesting set of experiments making use of a humanized mouse model to examine B cell co-infection with EBV and HIV-1 and its potential impact on both EBV and HIV-1 disease and latency. AIDS patients are particularly susceptible to EBV-associated lymphomas (due to a loss of EBV-specific CD4 T cells) and B cells have been proposed as an additional reservoir for latent HIV-1 (there is previous evidence of B cell infection by HIV-1 and of EBV positive cell line infection by HIV-1). The detailed in vivo co-infection studies here therefore have a basis and are important for our understanding of both viruses and their interactions. The manuscript investigates co-infection in a good model environment and the technical quality of experiments is excellent. My concerns lie with interpretation of some of the data and its potential significance and a lack of comparison with previous data available.

Major points:

1. A key question addressed was whether EBV infection (which activates B cells) makes B cells more susceptible to HIV-1 infection and if so why. Previous studies have shown that HIV-1 infection of B cells is enhanced when they are activated (e.g. by IL-4 and CD40 ligand). IL-4 and CD40L activation of B cells was previously shown to upregulate CD4 and CXCR4 but not CCR5 and the susceptibility of these cells to HIV-1 infection correlated with the level of receptor expression and was blocked by antibodies/peptides to CD4/CXCR4 (ref 14). B-cell infection can however also be blocked by antibodies against CD21 (CR2) or CD35 (ref 13 and 18). The authors show here that CXCR4-tropic strains and not CCR5-tropic strains infect EBV infected B cells and not CD19+ B cells and correlate this with CD4 and CXCR4 co-expression (Fig 1D). They propose that increased CD4 expression on EBV infected B cells is responsible (uninfected CD19+ B cells express high levels of CXCR4 but not CD4 by FACS and EBV infected cells actually downregulate CXCR4). The CD4 high and low B cell population experiments add some weight to this (Fig S1E and F), but the other candidate receptor/co-receptors (CR2 and CD35) are not incorporated into analysis. Importantly, published transcriptomics data indicate that CD4 is not always upregulated by EBV infection. These previous data are not referred to here and this is an important point. In the microarray study of Smith et al (2013, PLoS One) CD4 is upregulated by IL-4/CD40L after 7 days and not by EBV infection. The data of Nikitin et al (Cell Host Microbe, 2010) show that EBV infection (early or late) actually reduces CD4 expression to some extent. In both published studies CXCR4 is downregulated by EBV and IL-4/CD40L as seen here (Fig 1C). On the other hand CR2 (the EBV receptor) is upregulated by both IL-4/CD40L and EBV. There may well be more recent publicly available RNA-seq data sets to examine too. Given these contradictions, the mechanism for the enhanced susceptibility of EBV-infected B cells to HIV-1 infections warrants further investigation/analysis with CR2 considered as another molecule involved.

2. The transmission experiment in Figure 4 is a critical one in determining whether B cells can be an HIV-1 reservoir. Purified CD19+ B cells from co-infected mice do not transmit HIV-1 to naïve hosts

(Fig 4E). Only CD19+ B cells from co-infected hosts previously treated with CD8 depleting antibody can do this. This result is confusing. Surely this shows that B cells (EBV-infected or not) do not act as an efficient reservoir for HIV-1? What proportion of the donor transplanted cells are EBV/HIV positive in these two experiments? Does this explain the result?

3. The evidence that HIV-1 infected LCLs are more susceptible to T cell killing than normal LCLs is not convincingly demonstrated. Fig 5D shows that there is a small percentage of p24+ (HIV-1 positive cells) and this is reduced when EBV-specific T cells are added (FACS example shows 4.2% to 2.9%). These are very low numbers making error margins high (there is no statistical testing in the chart on the right). Is this significant? Where is the percentage killing of single EBV-infected cells shown for comparison to allow this increased susceptibility to be assessed?

Minor points:

1. Figure 5B should show up and downregulated GO terms for each comparison of data sets not just up for one and down for the other.
2. The two blue colors on the pie chart in Fig 2A are hard to distinguish so should be changed.
3. Coloring and shading in Fig S3D pie charts is very hard to decipher and a better color/shading scheme should be used.

Reviewer #3 (Comments to the Authors (Required)):

Coinfection of B lymphocytes with EBV and HIV-1 is an important problem for two reasons: first, because HIV-1 increases the risk of EBV-driven B-cell lymphoma, and second because persistence of HIV-1 in another cell type (other than CD4+ T cells) increases the difficulty in eradicating the reservoir of HIV-1 that persists during anti-retroviral treatment.

In the present manuscript, the authors show that X4-tropic HIV-1 (but not R5-tropic HIV-1) can productively infect EBV transformed B cells - lymphoblastoid cell lines (LCLs).

- The HIV provirus is integrated in the B cell genome, with the characteristics typical of HIV infection in T cells, i.e. in active genes, and with a weaker preference for integration in a primary DNA sequence motif.
- In NSG human mice, there was expansion of the CD8+ T cell population, but not CD4+ T cells, in animals coinfecting with EBV and HIV.
- EBV-specific CD8+ T cells responded less efficiently with production of IFN γ in the dually-infected humanized mice, and these mice developed macroscopic tumours more frequently than those infected with EBV alone.
- In apparent contradiction to the last observation, the authors observed that B cells taken from dually-infected mice and transferred to littermates that were reconstituted with the same preparation of human CD34+ cells could transfer HIV-1 infection, but only if the donor had been depleted in vivo of CD8+ T cells before the cells were isolated. Second, the dually-infected B cells were susceptible in vitro to killing by either EBV-specific or HIV-specific CD8+ T cell clones. Thus, although production of IFN γ seemed less efficient, the CD8+ T cells were able to recognize and kill the respective virus-infected cells.

Comments

The manuscript is clearly written and the results in principle support the main conclusions drawn by the authors. Although infection of B cells with HIV-1 and coinfection of B cells with EBV and HIV-1, have been observed for many years, as the authors acknowledge, the present findings do add to existing knowledge of this coinfection. However, the authors should also acknowledge more clearly some of the significant limitations of the work reported here, as follows.

1) B95-8 has a 10kb deletion in the genome, and is particularly efficient in transforming B cells. Indeed, this is why the strain is the standard strain used to produce LCLs. However, the behavior of the infected B cell differs from that in B cells infected with wild type EBV: the infection leads to initial transformation of the B cell, which then migrates to the germinal centre, where it differentiates to a resting memory state. In contrast, cells infected with B95-8 remain persistently activated. This is likely to change fundamental features of the co-infection with HIV-1, which in consequence are likely to differ from that in cells naturally infected with wild-type EBV and HIV.

2) The authors show that HIV-1 infection can persist in the EBV-infected B cells in the humanized mice, although this infection is controlled to an extent by the CD8⁺ T cell response. However, as the authors are well aware, even without the CD8⁺ T cell depletion, the humanized mouse model used does not fully and faithfully recapitulate the functioning (and still less the spatial aspects) of the human immune system. Together with the persistent activation of the EBV-transformed B cells studied here, it is therefore difficult to infer what these observations might mean in human infection.

Additional comments

1) It has recently become clear that Type 2 EBV can infect T cells. It would be interesting to study coinfection of T cells with this wild type EBV and HIV-1.

2) It would also be interesting to know whether there is selective infection with HIV-1 of EBV-specific B cells, and finally whether cell-to-cell contact is needed for this infection, i.e. a virological synapse.

McHugh, Myburgh et al., LSA-2020-00640-T

Point-by-Point response

The Reviewer's remarks and questions are in Black italic font.
Authors' remarks and answers are in Blue regular font.

We, the Authors, would like to thank all reviewers for their detailed review and constructive critique of our work. We have attempted to address all concerns of each reviewer and have implemented suggested changes to the manuscript text and figures. We believe this has improved the manuscript overall and will benefit the reader.

Reviewer #1 (Comments to the Authors (Required)):

In this manuscript, the authors investigated an unexpected change in cellular tropism of HIV1 in the context of Epstein Barr virus (EBV) co-infection. Notably, the authors showed that showed EBV-transformed B cells started to upregulate the HIV-coreceptors CD4 and CXCR4 and became susceptible to X4-tropic HIV infection in vitro. McHugh and colleagues subsequently confirmed a similar profile of proviral integrants in LCL and CD4 T cells. To confirm their observations in vivo, the authors took advantage of the only challenge model for HIV and EBV, namely human CD34+ HSC injected NSG mice which engrafted with components of a human immune system. As previously reported and consistent with clinical observations HIV infection reduced the numbers of CD4 helper T cells but not of CD8 T cells. McHugh et al provide evidence that CD8+ T cells expand in numbers upon co-infection but fail to control EBV in dually infected animals. Interestingly, transfer of B cells extracted from HIV/EBV co-infected humanized mice resulted in HIV infection in naive animals. While there is currently no evidence from the clinical literature suggesting that B cells from EBV/HIV co-infected patients are actually sensitive to HIV infection clinically, the data presented here are certainly intriguing and provide a strong rationale for more mechanistic follow-up studies. Overall, the manuscript is well written, the experiments are will designed and the data largely support the claims by the authors.

We thank the reviewer for the expert assessment of our manuscript and for acknowledging the intriguing nature of the results and their potential for spurring further follow-up investigations in a clinical setting.

Additional comments.

- Please apply stringent statistical analysis to all data presented to affirm significance.

We have now applied statistical analysis to all relevant comparisons of all data presented.

Do the authors have any data that would possibly hint a mechanism for the upregulation of CD4/CXCR4 in LCLs?

This is a very interesting question. First, we must clarify that we do not see up-regulation of CXCR4, in fact it is down-regulated on LCLs in comparison to the very high levels observed on EBV⁻ PBMC B cells (Fig. 1C) and we mention this in the text: "In line with the selective susceptibility of LCLs to X4-tropic HIV-1 infection, we found that a large fraction of the LCLs expressed CD4 on their surface and retained expression of CXCR4 at lower levels upon EBV transformation, whereas transcript levels of CCR5 were very low in selectively sequenced LCLs as previously reported (Fig 1C and Fig S1D) [31-33]."

Others and we have seen that B cells already express high levels of CXCR4 on their surface and during transformation by EBV tend to downregulate CXCR4 (Figure 1C, [1, 2]). Nevertheless, we believe it is still enough surface expression to be exploited by X4-tropic HIV-1 for entry in combination with CD4. Indeed we see that the combined expression of CD4 and CXCR4 correlated with the overall level of CXCR4-tropic HIV-1 replication in LCLs (Fig. 1D).

We have two possible mechanisms for the CD4 upregulation that we demonstrate in Fig. 1C-F. The first suggests a viral regulation of CD4 as a consequence of EBV nuclear antigen (EBNA) expression, specifically EBNA3A, B and C. The second mechanism might involve the expression of transcription factors in EBV infected B cells that are associated with CD4⁺ T cell differentiation during normal T cell development which may support aberrant CD4 expression.

A. A possible role for EBNA3A-C in CD4 regulation

In White et al. [3] the authors infected the EBV-negative Burkitt's lymphoma cell line BL31 with a number of mutant viruses and their revertants (revertant = the deletion was replaced with wild-type sequence as a control for second site mutations). The mutant viruses were generated in the context of the B95-8 EBV strain and consisted of independent deletions of all but the first few base pairs of EBNA3A (3AKO) and EBNA3B (3BKO) and a virus deleted for EBNA3C exon2 (3CKO) [4]. Additionally a mutant lacking the entire EBNA3 locus (E3KO) was created, thus lacking EBNA3A, B and C[3]. Transcriptome data was acquired from multiple, stable BL31 cell cultures upon infection with the individual mutant, revertant or wild-type virus via RNA Affymetrix Exon microarray. Because BL31 cells are an established cells line, this approach allows the investigation of host gene regulation independent of efficient immortalization, for which EBNA3A and C are indispensable.

In this analysis the EBV WT and revertants induced CD4 expression in the BL31 cell line whereas all the individual EBNA3 mutants did not. Additionally, the EBNA3-locus KO virus also failed to induce CD4 expression in BL31 cells. Please find the results of this analysis below in Table R1 and Figure R1. Taken together this suggests that the EBNA3 proteins may act cooperatively in upregulating CD4 mRNA. Cooperative regulation of host gene expression by the EBNA3 proteins has been described before [3] and is thus not an unlikely mechanism for CD4 upregulation which should be investigated in depth in future studies. We also would not like to exclude the possibility of other EBV genes playing a role in CD4 up regulation. In both LCLs and in humanized mice the predominant gene expression pattern by far is Latency III and thus these EBV-infected B cells express the full repertoire of EBV latent gene, while lytic activity can also always be detected albeit at lower levels[5].

Host gene	uninfected BL31 vs WT		3AKO vs WT		3BKO vs WT		3CKO vs WT		E3KO vs WT	
	p	Fold-Change	p	Fold-Change	p	Fold-Change	p	Fold-Change	p	Fold-Change
CD4	3.0 ⁻⁶	-11.0	6.8 ⁸	-14.3	2.7 ⁻⁸	-16.9	5.04 ⁻⁸	-15.0	4.6 ⁻⁷	-15.1

Table R1 This table contains data derived from White et al. [3] Table S1 containing a complete list of all genes with contrast statistics showing p-value (p ANOVA) and fold change of mutant EBV-infected cell lines as compared to wtBAC-infected and uninfected BL31 compared to wtBAC. Negative fold change represents a lower expression level in mutant compared to wtBAC cells indicative of a role of the respective viral nuclear antigen(s) in inducing host gene expression.

[Editorial Staff has removed Figure per Authors' Request]

B. A possible role for transcription factors typically involved in helper T cell differentiation during CD4 expression in EBV infected B cells.

The zinc finger and BTB domain containing 7b (*Zbtb7b*) gene, encodes a central transcription factor for CD4⁺ T cell development, called T_H inducing POZ-Kruppel factor (ThPOK, reviewed in [6]). A point mutation in *Zbtb7b* was found to be responsible for the absence of a CD4⁺ T cells in the periphery in a natural mutant mouse strain termed helper deficient mice [7]. In helper deficient mice, positive selection of thymocytes via MHC class II-restricted TCRs is intact, however, differentiation is re-directed into the CD8⁺ T cell lineage. Ectopic expression of ThPOK via stable transgenesis or retroviral transduction not only rescued the phenotype but also induced re-directed differentiation of MHC class I-selected thymocytes into CD4⁺ T cells. Expression of *Zbtb7b* in post-selected thymocytes is thus both essential and sufficient to commit to the CD4⁺CD8⁻ lineage in T cells. We analyzed a previously published RNAseq dataset of gene expression during EBV-infection of B cells from the group of Wolfgang Hammerschmidt [1]. We found that *Zbtb7b* was already highly expressed and tended to increase during EBV mediated transformation of B cells (Figure R2).

Gata3 is another transcription factor involved in CD4⁺ T cell development in general but may also be involved in *Zbtb7b* expression specifically [6]. Conditional inactivation of *Gata3* in transgenic mice using a Cre-lox system dramatically reduced the development of CD4 single positive thymocytes [8]. *Gata3* binds regions in the *Zbtb7b* locus and *Gata3*-deficient thymi lack expression of ThPOK (*Zbtb7b* gene) indicating that that *Gata3* may be necessary for *Zbtb7b* expression [9]. Transgenic expression of ThPOK failed to rescue CD4 single positive thymocyte development caused by *Gata3* deficiency [9], indicating that *Gata3* mediates both a ThPOK-dependant and -independent pathway for driving CD4⁺ T-cell development. In the previously mentioned dataset from Mrozek-Gorska et al. [1] we found *Gata3* to be highly upregulated and expressed at progressively higher levels over time after day 4 upon B cell infection with EBV (Figure R2).

Myb is another transcription factor involved in promoting CD4⁺ T cell lineage development [10]. Although conditional knockout of *Myb* in double positive thymocytes resulted in a decrease of CD4 single positive and an increase in CD8 single positive cells, redirection of MHC class II-restricted cells into CD8 single positive cells was not observed. Of note, *Myb* binds to the *Gata3* locus and thus may induce *Gata3* expression. In line with this notion the data from Mrozek-Gorska et al. suggests that *Myb* expression is rapidly induced to high levels on day 3 before *Gata3* expression, were levels remain until day 15 post infection (Figure R2).

In summary, the transcription factors involved in CD4⁺ T cell lineage commitment during normal T cell development are expressed and even induced in EBV infected B cells including ***Zbtb7b***, which is necessary and sufficient for CD4 single positive thymocyte generation; ***Gata3*** which is thought to be involved in both *Zbtb7b* expression and to directly drive CD4⁺ T cell development and ***Myb***, which is also known to promote helper cell development possibly via increasing *Gata3* expression. Furthermore, both *Myb* and *GATA3* have also been identified as important factors in T cell development before helper cell commitment (i.e. at the double negative stage) [6].

In addition to the two possible mechanisms of CD4 expression after EBV infection of B cells, as outlined above, which are not mutually exclusive possibilities, we also think it is possible that the expression of CD4 is a stochastic event. We speculate that B cells that would, under normal circumstances, start to exhibit non-cell type

specific protein expression would be more likely to undergo apoptosis, but these cells exhibiting lineage promiscuity are rescued from this more typical fate by the many anti-apoptotic properties of EBV itself.

In summary, we do have first indications regarding the mechanisms of up-regulation of CD4 in infected B cells directly or indirectly via EBV infection. However, we feel the inclusion of this rather speculative data in the manuscript would be premature without additional experimental validation and is not the main focus of this manuscript (i.e. HIV-1 infection as a consequence of CD4 expression).

[Editorial Staff has removed Figure per Authors' Request]

Have the authors analyzed other (cytokine) receptor loci to ascertain how specific the change in transcription is to CD4/CXCR4?

We present here for the reviewer an analysis of cytokine and chemokine receptor gene expression during EBV transformation from Mrozek-Gorska et al. [1] (Fig. R3). The list of 76 receptors was based on the list of cytokines and chemokines published by the Madame Curie Bioscience Database curated by Mark J. Cameron and David J. Kelvin (<https://www.ncbi.nlm.nih.gov/books/NBK6294/>). Only common cytokines and chemokines with recognized immune functions and the most common hematopoietins were included, whereby other “growth factors”, neurobiological proteins and “trophins” were excluded. Days 0 and 1 were excluded from the analysis since at these time points the likelihood of non-B cell contamination within the putative B cell population is high. These contaminating cells will die over the course of the EBV immortalization of B cells in culture and may give the false impression of a reduction of certain receptors within B cells at later timepoints. This is not ideal as exclusion of early timepoints may on the other hand obfuscate true rapid downregulation of receptor gene expression in B cells. We could, however, within this overview of (cyto-/chemokine) receptor gene expression find expected downregulated genes such as CXCR4 in cluster 1 and upregulated genes such as CD4 (CD4 is the IL-16 receptor) and TNFRSF8 in cluster 3. Interestingly, TNFRSF8 encodes for CD30, which we have recently published on as a relatively specific receptor upregulated during EBV infection of B cells *in vivo* in pediatric EBV+ PTLD patients, in EBV infected humanized mice that developed EBV tumor formation under immunosuppression and expressed *in vitro* in LCLs [11]. Overall, cluster 1 contained 25 receptor genes that generally trended downward, cluster 3 contained 34 genes and trended upward during transformation, whereas cluster 2 genes (n=16) showed either transient regulation - which could also be indicative of slight variance from day to day at very low expression levels - or late up regulation of expression.

As one would expect, EBV transformation brings much disruption to the normal B cell transcriptome. It is thus not surprising that this general change in transcriptional activity extends to cytokine and chemokine receptors and results in both up- and down-regulation of many of these transcripts. We can as such not conclude that CD4 upregulation is specific among cytokine/chemokine receptors since many other receptors are also upregulated. We hope this has nonetheless shed some light on the reviewer's question.

[Editorial Staff has removed Figure per Authors' Request]

- Do the transformed B cells acquire any other markers of T helper cells (phenotypic or functional)?

This is a simple and interesting question, but in reality very hard to answer. In this study we did not investigate CD4 expression because of its attribute as a T cell subset defining marker, but rather because of its use by HIV as an entry co-receptor. The fact that it defines T helper cells within a CD3 positive lymphocyte population is from virological perspective entirely incidental. Nevertheless, we are also intrigued by the reviewer's idea and have attempted to address this inquiry as comprehensively as possible.

In order to define a meaningful starting point we used the set of 198 genes identified by Abbas et al. [12] that were up-regulated when comparing CD4⁺ T cells and B cells via microarray as this gives us a likely indication of positive CD4⁺ T cell gene expression relative to B cells before EBV mediated changes. All genes within this set were then plotted over time during EBV transformation of B cells derived from the previously published RNAseq data by Mrozek-Gorska et al. [1]. Days 0, 1 and 2 were excluded from the analysis since at these time points the likelihood of CD4⁺ T cell contamination within the putative B cell population is still high. For a more detailed explanation and data supporting this assumption please see our response to Reviewer 2 point 1 and Figure R5. Hierarchical clustering was then performed to group genes that responded similarly during the course of the EBV transformation process (Figure R4). Genes in cluster 1 tended to decrease during transformation, these included T cell related genes such as CD7, CD3D, CD247 (=CD3zeta) possibly initially derived from residual T cells at the beginning of the transformation process, which die and are lost over time in the culture. Clusters 2, 3 and 4 contain genes that are either not clearly regulated in one direction or the other during the transformation or were only transiently up-or down regulated. Some of these genes had very low expression overall, as such the extent of their transient dysregulation should not be overemphasized (ELANE for example was not expressed at all; Cluster 3). Finally cluster 5 contained 58 of the total 197 genes that tended to increase over time including CD4, CD27, CD28 and GATA3. In summary, we cannot conclude that EBV transformed B cells acquire a global expression pattern of T helper cell like gene expression. Only around a quarter of the genes that are high in CD4⁺ T cells compared to B cells, trend upwards during EBV transformation while the remaining are unchanged, only transiently upregulated or trend downward. Furthermore, many of the genes in the GSE22886 set are not exclusive to CD4⁺ T cells. CD27 for example is a marker expressed in memory B cells and EBV is known to drive differentiation of infected naïve B cells toward this B cell subset.

CD4 is also expressed by antigen presenting cell (APC) subsets, including human monocytes and macrophages, and is the receptor for IL-16, a cytokine with chemoattractive properties. In [13] MACS sorted CD4⁺ and CD4⁻LCLs were tested regarding their immunostimulatory capacity by assessing their ability to induce

proliferation of CD4⁺ T cells, CD8⁺ T cells or CD56⁺ NK cells in a mixed lymphocyte reaction. No difference in immunostimulatory capacity was seen. However, CD4⁺ LCLs showed a small but significant increase in IL-16 dependent migratory activity indicating that CD4 expression is at least partially functional on LCLs as the IL-16 receptor [13].

[Editorial Staff has removed Figure per Authors' Request]

- Figure 1

A. Please provide the data of at least 2 donors

We have included the data of an additional donor as an additional panel in Figure 1A.

B. This reviewer is wondering why the kinetics of p24 differ between human donor-derived LCLs and NSG-HIS mice derived LCLs. This indicates their humanized mice does not mimic the actual human behavior. Please discuss and provide the data of two additional mice with RNA copy on 20, 25 days post infection.

We agree that the overall and maximum HIV-1 replication differs between different LCLs, we believe this can be explained by the relative surface levels of entry receptors (CXCR4 and CD4 in Figure 1D). However, we disagree that there is any discernable difference between LCLs that can be attributed to their source (human vs. huNSG). In Figure 1B the mouse derived LCLs seem to replicate at a level that is lower than 2 of the human donor-, but higher than 2 further human donor-derived LCLs. Human donor-derived LCLs were transformed *in vitro*, humanized mouse derived LCLs were transformed *in vivo*. One could argue that *in vivo* infection, transformation and establishment of immortalized B cells within a milieu of immune competent and expanding T cells is even more representative of the actual human behavior. In any case, culturing the cells for longer than 2 weeks for p24 antigen quantification in the supernatant is associated with some difficulties as the decline in p24 observed in some samples towards the end of the assay is likely due to impairment of the viability of the cells as they continue to proliferate and crowd the well. At this point the assay is usually stopped as splitting cells or changing the media would no longer allow for meaningful evaluation of the protein results from the supernatant. Is this the difference in kinetics the reviewers is referring to? LCLs are primary cells and we observe large variation in growth and phenotype even among multiple transformation experiments from the same donor cells. As such we would caution against overinterpretation of small differences seen between different LCLs, especially toward the end of the HIV-1 infection experiments *in vitro*.

We could, unfortunately, due to the COVID-19 lockdown in our country not perform the additional experiments and analysis suggested by the reviewer. However, we would like to point out that we present HIV-1 (NL4-3) infection *in vitro* of two additional humanized mouse derived LCLs from *in vivo* infection with the M81 EBV strain in Figure S1A.

- Fig.2 There is a considerable difference in the number of samples (CD4+ T cells vs LCLs) that were included in this analysis. Why are the numbers so heavily skewed towards LCLs? Also, it would have been desirable to compare the integration profile of LCL with CD4+high and CD4+low expressing cells.

In Figure 2A the “n = ” indicates the number of integration sites identified via nrLAM-PCR, sequencing and mapping with the InStAP pipeline and not the number of cells or samples analyzed. We have amended the figure legend to make this clearer to the reader:

“(A) Distribution of HIV-1 integration sites in the host genome comparing CD4+ T cells and autologous LCLs 2 days post in vitro infection with NL4-3 (n = number of integration sites).”

Although we used the same amount of cells at infection we happened to identify more integration sites in the LCLs. This is likely a technical reason as we recovered more DNA from LCL samples to begin with since LCLs seem to be more resilient to vitro culture even for very short periods. However, the difference in the number of identified integration sites for each source cell type does not skew our interpretation of the comparative characterization of HIV-1 integration in LCLs and T cells. The integration sites are still similar with regard to the nucleotide sequence upstream of the HIV-1 integration sites and to the rate of active transcription in integrated transcriptional units. Furthermore, our integration site analysis is in line with previously published integration site characterization of T cells as stated in the main text.

We agree that the investigation of CD4^{high} and CD4^{low} LCLs could be interesting. However, in a first step we found that the comparison between X4-tropic HIV-1-infected LCLs and T cells would be the most informative regarding the biology of HIV-1 integration. Our reasoning was that the investigation of CD4 low cells would have likely resulted in a lower frequency of integration sites to characterize and as such it was not our first priority and we opted to compare CD4^{high} LCLs with CD4+ T cells.

- Fig.3B. The reviewer is not convinced that CD8+ T cell frequencies are actually decreased during co-infection based on the histological data provided. Please provide quantitative data.

We agree with the reviewer that CD8+ T cells are not decreased during co-infection compared to EBV-single infection. For clarity we have rewritten the corresponding results section to the following: “Conventional H&E and immunohistochemistry (IHC) staining of splenic sections from dual-infected animals revealed the presence of tumor-like lesions containing CD20+ B cells with a high frequency of EBNA2-positive cells and similar CD8+ T cell infiltration compared to EBV-infected mice (Fig 3B).”

Furthermore, we mention explicitly, when referring to the quantitative data of blood in Figure 3C, that there is a similar CD8+ T cells expansion between afore mentioned groups. Moreover, we demonstrate the similarity in CD8+ memory T cell subsets expansion between EBV and EBV/HIV infected animals in the quantitative data of the spleen in Figure S2C and the similarity in overall splenic CD8+ T cell expansion in Figure S2G. Panel S2G now also includes the data from splenic CD4+ T cells. Our conclusion from these data presented in Figure 3 and S2 is that CD8+ T cells clearly expand to a similar degree in dual infected humanized mice compared to EBV-single infected animals and are not decreased.

- Fig.5

o The data of "HIV-infected LCL vs non-infected LCLs" is not so helpful. LCLs should be stratified by expression levels of CD4+ since the phenotype of "bulk LCLs" is dissimilar to actual CD4+T cells.

o What was the rationale for not comparing HIV-infected LCL (CD4+high) and non-infected LCL (CD4+ high)?

This was actually performed as stated by the reviewer. We employed CD4⁺ high LCLs for the analysis of integration sites and the relevant section of the methods ("*HIV-1 integration site analysis*") has been amended to include this fact.

The rationale behind this was, as the reviewer hints at, to increase the amount of infected LCLs as much as possible in order to 1) increase the amount of integration sites for analysis and 2) increase the likelihood of seeing HIV-1 induced changes on the RNA level compared to non-infected LCLs at 2 days post infection.

Minor comments:

1. Page 13: Please add "in vitro" after the sentence "Furthermore, we found that dual-infected mice..."

We have amended the sentence in question. It now reads: "Furthermore, we found that dual-infected cells are highly susceptible to immune control via CD8⁺ T cells *in vitro* and may only contribute to HIV-1 viremia when CD8⁺ T cell function is severely impaired."

Reviewer #2 (Comments to the Authors (Required)):

The manuscript by McHugh and Myburgh and colleagues describes an interesting set of experiments making use of a humanized mouse model to examine B cell co-infection with EBV and HIV-1 and its potential impact on both EBV and HIV-1 disease and latency. AIDS patients are particularly susceptible to EBV-associated lymphomas (due to a loss of EBV-specific CD4 T cells) and B cells have been proposed as an additional reservoir for latent HIV-1 (there is previous evidence of B cell infection by HIV-1 and of EBV positive cell line infection by HIV-1). The detailed in vivo co-infection studies here therefore have a basis and are important for our understanding of both viruses and their interactions. The manuscript investigates co-infection in a good model environment and the technical quality of experiments is excellent. My concerns lie with interpretation of some of the data and its potential significance and a lack of comparison with previous data available.

We thank the reviewer for highlighting the importance of our investigations for the understanding of EBV HIV-1 interactions and for remarking on the technical quality of the experiments. We have attempted to address the remaining concerns regarding the interpretation of our data and additional comparisons to previously published data below.

Major points:

1. A key question addressed was whether EBV infection (which activates B cells) makes B cells more susceptible to HIV-1 infection and if so why. Previous studies have shown that HIV-1 infection of B cells is enhanced when they are activated (e.g. by IL-4 and CD40 ligand). IL-4 and CD40L activation of B cells was previously shown to upregulate CD4 and CXCR4 but not CCR5 and the susceptibility of these cells to HIV-1 infection correlated with the level of receptor expression and was blocked by antibodies/peptides to CD4/CXCR4 (ref 14). B-cell infection can however also be blocked by antibodies against CD21 (CR2) or CD35 (ref 13 and 18). The authors show here that CXCR4-tropic strains and not CCR5-tropic strains infect EBV infected B cells and not CD19+ B cells and correlate this with CD4 and CXCR4 co-expression (Fig 1D). They propose that increased CD4 expression on EBV infected B cells is responsible (uninfected CD19+ B cells express high levels of CXCR4 but not CD4 by FACS and EBV infected cells actually downregulate CXCR4). The CD4 high and low B cell population experiments add some weight to this (Fig S1E and F), but the other candidate receptor/co-receptors (CR2 and CD35) are not incorporated into analysis. Importantly, published transcriptomics data indicate that CD4 is not always upregulated by EBV infection. These previous data are not referred to here and this is an important point. In the microarray study of Smith et al (2013, PLoS One) CD4 is upregulated by IL-4/CD40L after 7 days and not by EBV infection. The data of Nikitin et al (Cell Host Microbe, 2010) show that EBV infection (early or late) actually reduces CD4 expression to some extent. In both published studies CXCR4 is downregulated by EBV and IL-4/CD40L as seen here (Fig 1C). On the other hand CR2 (the EBV receptor) is upregulated by both IL-4/CD40L and EBV. There may well be more recent publicly available RNA-seq data sets to examine too. Given these contradictions, the mechanism for the enhanced susceptibility of EBV-infected B cells to HIV-1 infections warrants further investigation/analysis with CR2 considered as another molecule involved.

We thank the reviewer for this comment and note that it contains two, related but distinct, main points which we will aim to address in succession below:

A) Published transcriptomics data do not always indicate CD4 upregulation in B cells upon EBV transformation.

B) CD21 and CD35 should be considered as candidate entry receptors.

A) Published transcriptomics data do not always indicate CD4 upregulation in B cells upon EBV transformation.

The reviewer mentioned “*The data of Nikitin et al (Cell Host Microbe, 2010) show that EBV infection (early or late) actually reduces CD4 expression to some extent.*” A possible downregulation of CD4 in B cells upon EBV infection would indeed be a major contradiction, as we had estimated the average frequency of CD4 protein surface expression via conventional flow cytometry to be less than 0.5% on B cells and in LCLs we and others have found positivity of up to 30% by conventional flow cytometry (This manuscript Figure 1, supplemental Figure 1 and [13]. If indeed CD4 mRNA would decrease upon EBV infection in B cells, what level of expression was it decreasing from?

To identify B cells and quantify the frequency of CD4⁺ expression with a different method we used the previously published mass cytometry (CyTOF) dataset from 15 healthy control subjects [14] publicly available from flow repository at <http://flowrepository.org/experiments/2166/> and performed FlowSOM clustering as performed by Galli et al. to identify B cells [14]. We demonstrate here for the reviewer in Figure R5A that with this method of single cell based determination of surface protein expression that $\leq 0.5\%$ of B cells express CD4 on their surface. The use of high-dimensional mass cytometry presents certain advantages. Through the use of a 7-chooses-3 CD45-live cell barcoding in the dataset we analyzed, the probability of artifacts due to cell doublets is exponentially reduced [15]. This is particularly important when characterizing small cell subsets. Combined with the automated analysis to define cell type, this provides an unbiased approach to characterize the expression of CD4 on B cells. Importantly, the results from this investigation are completely in line with results from our conventional flow cytometry approach. CD4 expression on the surface of non-EBV infected B cells was below 0.5% measured by both methods.

In the report from Hoennscheidt et al. [13] the authors evaluated CD4 protein and RNA expression in 20 LCLs derived from 14 donors upon transformation with the B95.8 strain (from six donor 2 two independent immortalizations were performed). In line with our investigations the number of CD4⁺ cells was different in different LCLs, but crucially, the expression of CD4 was detected at over 10% on the surface of all investigated cell lines by flow cytometry.

How then could it be possible that this is not reflected for example in the transcriptome data from Nikitin et al. [2]? The authors of the paper compared the transcriptomes of bulk B cell populations, EBV-infected B cells at 7 days post EBV infection and transformed B cells (LCLs). They report in the supplemental experimental procedures that within the purified B cell populations “*Purity was routinely greater than 90% as determined by flow cytometry*”. This means the remaining 10% were likely composed of non-B cells with a composition typical of healthy human adult PBMCs. As such this population likely contains a non-negligible CD4⁺ T cell fraction, which precludes this data’s utility in assessing discrete CD4 up- or down regulation. We should point out that this data set was not created with the goal of detecting small changes in protein expression on individual cells, rather to investigate the large global changes in the transcripts during early and late EBV

infection of the most significant and meaningful magnitude which they further analyzed for enrichment of GO terms. They found that the global gene expression analysis supported their previous investigations of the existence of a hyper-proliferative period and DNA damage response early after infection that is later attenuated upon completed transformation. We find this dataset is still valid for detecting the pronounced and dramatic CXCR4 downregulation also observed by us on the protein level and we now cite this paper as support for this finding in the results section.

In line with the idea that apparent down regulation of CD4 RNA expression in EBV-infected B cells is indeed an unfortunate artifact of the experimental setup in bulk transcriptome studies, we present here in Figure R5B for the reviewer an analysis of the time resolved RNAseq analysis from data by Mrozek-Gorska et al. [1]. Here the authors used B cells purified from tonsils and analyzed the RNA expression profile over time during EBV-infection up to 14 days post infection. We can observe that CD4 RNA dips during the first 3 days of in vitro infection, then increases again during the following 5 days and seems to remain stable up to day 14 whereby a larger variance in the data is observed. The dip in CD4 RNA expression during the first days of infection is paralleled by a reduction of gene expression of the T cell receptor complex and T cell lineage antigen CD3, specifically CD3G, CD3D, CD274 (CD3zeta chain) follow this same pattern. This is clearly indicative of CD4⁺ T cell contamination of purified B cell populations at the beginning of such experiments that die in culture over time during EBV mediated transformation and rescue of infected B cells. This may explain why in similar data sets, like those published by Nikitin et al. [2] or Smith et al. [16], no change or even a reduction of CD4 may be erroneously suspected after EBV infection of B cells.

The reviewer states correctly that *"In the microarray study of Smith et al. (2013, PLoS One) CD4 is upregulated by IL-4/CD40L after 7 days..."*, however this CD4 upregulation is also associated with an upregulation of the T cell specific genes CD3G, CD8A indicating a possible unspecific expansion of contaminating T cells in the IL-4/CD40L bulk analyzed cell population (Supporting Information Datafile S2 of Smith et al. [16]). Though cell composition is not reported, this notion is further supported by lower expression of CD3G, CD3E, CD3D when comparing EBV infected B cells for 7 days vs IL4-4/CD40L treated cells, making this analysis difficult to utilize for assessing the effect of IL4/CD40L or EBV infection on CD4 expression. Again we would like to point out that this does not invalidate this or similar datasets for the purposes of their respective investigation in the context of specific validation experiments relative to observed changes in global B cell gene expression.

In summary, transcriptomics from bulk cell populations are most useful to screen for sizable differences or global changes in expression patterns but are less adequate approaches to disprove small or medium sized changes in protein expression on the surface of cells that can be identified via single cell approaches such as flow cytometry. We conclude that previously published datasets do not contradict our findings but rather support enhanced CD4 protein expression on the surface of EBV transformed B cells such as the study by Hoennscheidt et al. [13] (please also see our answer to Reviewer 1 question 2 regarding possible mechanisms for CD4 upregulation during EBV infection).

[Editorial Staff has removed Figure per Authors' Request]

B) CD21 and CD35 should be considered as candidate entry receptors.

One of the most compelling reasons CR2 and CR1 molecules likely cannot substitute for CD4 and co-(CCR5 or CXCR4) mediated entry during HIV-1 entry of LCLs is that R5-tropic viruses (YU-2, JR-CSF) never replicate in LCLs cultures. To date we've found no evidence that R5-tropic viruses show any signs of HIV-1 splice-specific gene expression, or replication via p24 protein accumulation in the supernatant in side by side comparisons with the same MOI of X4 or X4/R5 dual tropic HIV-1 strains (Figure 1 and S1).

Regarding the two publications by Gras et al. mentioned by the reviewer: In 1991, Gras and Dormont describe complement mediated antibody dependent enhancement of HIV-1 infection of a single EBV transformed cell line (LCL, IC.1) [17]. For these investigations the authors used the HIV-1 LAV-1 strain, which in retrospect happened to be an X4-tropic strain. A fortunate happenstance as the identification of CCR5 and CXCR4 as major co-receptors required for entry of HIV-1 alongside CD4 was only years later [18-21]. At the time CD4 expression on the surface was not detectable by flow cytometry. However, the authors acknowledge that based on their experiments "CD4 is strictly required for infection" [with HIV-1] whereas "blockade of CR2 inhibited the enhancement mechanism" [17].

In 1993, the authors investigated purified B cell populations, which were PMA stimulated for 2 days [22]. Without prior PMA stimulation p24 levels in the supernatant were just slightly above background levels of the assay (Figure 4 of Gras et al. [22]). Infection of B cells under these conditions depended completely on preincubation of HIV-1 with HIV-positive serum and complement sufficient normal serum. Across multiple experiments this led to a transient p24 production measured in the supernatant around day 3 or 4 which subsided by day 6 p.i. (see esp. Figure 2B of Gras et al. [22]). The LAI strain of HIV-1 was used and it is X4-tropic. CXCR4 was not assessed in this study, again, this was not a known concept at the time. The blocking experiments in the serum should be viewed with some additional caution in our opinion since blocking antibodies toward CD4 and CD19 seem to lower p24 production to a similar degree as CD21 and CD35 under conditions with HIV specific serum (Figure 8A of Gras et al. [22]). Antibodies against CD4 and CD19 were used at 0.5µg/ml whereas antibodies against CD21 and CD35 were used at 1 µg/ml. Under conditions without HIV-specific serum preincubation of B cells, in our view, CD4 blocking had a more robust effect on p24 levels compared to CD21 or CD35 blocking (Figure 8A: last two columns within each group, Gras et al. [22]) though it is hard to interpret as statistical testing was not performed. Figure 8B depicts results representative of a singular experiment. B cells contain high levels of CD21 (CR2) this also makes it the target cell for EBV infection. Under conditions of HIV-specific serum preincubation of HIV-1 the authors add antibodies against CD21 and CD35 at 4µg/ml each to the cells which may also result in an unspecific steric hindrance, this was not controlled for adequately and should in our opinion not be overinterpreted. We do not agree with the interpretation that CD21 and CD35 are entry receptors for HIV-1 in B cells.

In contrast to the above studies in our investigations of HIV-1 infection of EBV transformed B cells (LCLs) no preincubation with human serum or complement factors were necessary to facilitate infection and infection correlated with CD4 and CXCR4 surface expression on the LCLs. HIV-1 replication was higher in CD4 positive sorted LCLs compared to CD4 negatively sorted counterparts and crucially infection was dependent on an X4 tropism of the HIV strains precluding to a large degree the possibility of an unspecific HIV-1 entry mechanism.

We present here for the reviewer in Figure R6 additional data on CD21 and CD35. Though there may be a transient upregulation of CR2 (CD21) and CR1 (CD35) gene expression post EBV infection of B cells in vitro, as mentioned by the reviewer, by day 14 of the transformation, transcripts of both receptors seemed lower compared to non-infected B cells (Figure R6A). This was mirrored in the lower CD21 and CD35 surface protein expression on LCLs compared to PBMC B cells (Figure R6B). In contrast, but similar to previously published findings, we demonstrate in Figure 1A that PBMC B cells cannot be infected with HIV.

We could unfortunately due to the COVID-19 lockdown in our country not perform HIV-1 live virus infections for correlation of viral replication with CD21 and CD35 surface expression at the same time. We used previously generated data of HIV-1 replication in LCLs and correlated it with RNA sequencing data obtained from these LCLs which was comparable in terms of these experiments. There did not seem to be a positive association between CR2 (CD21) or CR1 (CD35) transcript levels and overall HIV-1 replication in LCLs in these data (Figure R6C). Though we can not currently rule out the possibility that CR1 and 2 have a modifying effect, we do not believe it is a main mechanism for HIV-1 entry of B cells once they have been transformed via EBV as the surface levels of CR2 and CR1 are generally lower compared to non-infected B cells. Our conclusion from our own data and the studies discussed above is that B cell infection with HIV-1 is limited to CXCR4-tropic strains and is CD4 dependent, whereby under specific conditions CD21 and CD35 may have modifying effects but are not directly receptors for HIV-1 entry.

To address the CD21/CD35 issue we now more precisely introduce the role of CD21 and CD35 in HIV-1 infection of B cells in the introduction as follows:

“In vitro, HIV-1 can infect IL-4 and/or CD40L-activated primary B cells due to the expression of CD4 and CXCR4 and infection may be enhanced in PMA activated B cells via binding of HIV-1 in immune complexes by complement receptors CD21 and CD35 [12-15].”

In the discussion we have noted the possible additional role that these surface molecules may play *in vivo*:

“As such, HIV-1 binding on the surface of B cells as immune complexes, as previously described for non-infected B cells [67], may only be poorly modeled in this system. Since EBV infected cells downregulate CD21, the extent to which EBV-infected B cells bind immune-complexed HIV-1 via CD21 and thus facilitate infection of susceptible cells, as described previously for non-EBV infected B cells will have to be further evaluated.”

[Editorial Staff has removed Figure per Authors' Request]

2. The transmission experiment in Figure 4 is a critical one in determining whether B cells can be an HIV-1 reservoir. Purified CD19+ B cells from co-infected mice do not transit HIV-1 to naïve hosts (Fig 4E). Only CD19+ B cells from co-infected hosts previously treated with CD8 depleting antibody can do this. This result is confusing. Surely this shows that B cells (EBV-infected or not) do not act as an efficient reservoir for HIV-1? What proportion of the donor transplanted cells are EBV/HIV positive in these two experiments? Does this explain the result?

This depends somewhat on how one wishes to define the term reservoir. Our interpretation relevant to these experiments would be: A cell type in which a replication-competent form of the virus accumulates and persists with more stable kinetic properties than in the main pool of actively replicating virus. From a practical standpoint this is frequently equated to cells from which HIV-1 can re-emerge upon cART cessation (both definitions adapted from [26]).

The transmission experiment in Figure 4 was not strictly designed to assess whether the dual infected cells are a HIV reservoir but rather provides evidence of productive HIV-1 replication in dual-infected B cells and thus transfer of autologous B cell populations can transfer both viral diseases to recipient virus naïve humanized mice. We have clarified this now in the text as follows:

“We aimed to determine whether in vivo EBV/HIV co-infected B cells actively replicate HIV-1 upon transfer to a HIV-1 naïve host.”

Indeed as the reviewer suggests, the higher number of dual-infected cells in donor mice with CD8 depleting antibody, as assessed histologically with the automated unbiased Vectra cell counting algorithm for the detection of EBNA2+ and p24+ cells (Figure 4C), explains the result that upon transfer of the B cell fractions only recipients of OKT8 treated donors developed HIV-1 viral loads. These results are also in line with more frequent detection of splice-specific HIV-1 transcripts via RT-qPCR in B cells fractions of mice that had received CD8 depleting antibody (Figure S3A). As such the results from the transmission experiment support the idea that these B cells are productively infected.

In general, productive infection is a prerequisite for reservoir potential. We at present provide only little evidence regarding dual-infected cells as a reservoir in the sense of the working definition above in Supplemental Figure 3C and D. The only human cells these cART treated mice received were the EBV transformed cells (LCLs) some of which were productively infected by HIV-1. As such the only cell HIV-1 could have sustained the cART treatment were LCLs. We agree that if B cells act as a reservoir they are likely less frequent and less efficient compared to canonical reservoir cells.

What we can say is that when these cells are detectable in a sizable proportion, which is the case when CD8 T cells are depleted, these cells can contribute to the development of a detectable HIV-1 burden. Again, these results indicate a surprisingly strong CD8 immune response toward HIV-1 infected B cells, which we assess experimentally with multiple EBV and HIV-1 specific CD8⁺ T cell clones in the following results paragraph, in Figure 5 and supplemental Figure S5.

We also mention these arguments against EBV infected B cells as a HIV-1 reservoir in the discussion as follows:

“Furthermore, we provide evidence that EBV-transformed cells may harbour HIV-1 in vivo after ART treatment, during which no peripheral HIV-1 RNA load can be detected. Dual-infection, however, is likely a rare event as it pertains solely to X4-

tropic viruses which occur in only 50% of patients at late stage disease [52] and only a fraction of the EBV-infected B cells express the requisite surface receptors CD4 and CXCR4. Furthermore, we found that dual-infected cells are highly susceptible to immune control via CD8⁺ T cells in vitro and may only contribute to HIV-1 viremia when CD8⁺ T cell function is severely impaired.”

3. The evidence that HIV-1 infected LCLs are more susceptible to T cell killing than normal LCLs is not convincingly demonstrated. Fig 5D shows that there is a small percentage of p24+ (HIV-1 positive cells) and this is reduced when EBV-specific T cells are added (FACS example shows 4.2% to 2.9%). These are very low numbers making error margins high (there is no statistical testing in the chart on the right). Is this significant? Where is the percentage killing of single EBV-infected cells shown for comparison to allow this increased susceptibility to be assessed?

We have now included the statistical test data as requested by the reviewer. The frequency of p24⁺ LCLs (after pre-gating on the lipophilic membrane dye with which the LCLs were stained) was compared in conditions with EBV specific T cell clones to corresponding control conditions without T cell clone addition and was assessed with a Wilcoxon matched pairs test. Though the frequencies of p24⁺ LCLs was low at times - as pointed out by the Reviewer - the effect of EBV specific clone addition was significant. HIV-1 specific clones as expected also led to significant and perhaps more pronounced specific depletion of HLA-matched HIV-1 infected LCLs compared to EBV specific clones whereas an Influenza specific CD8⁺ T cell clone did not specifically target HLA matched dual infected cells (Supplemental Figure S5).

In Figure 5D the rate of single EBV infected killing is considered as the analysis does not only deal with the number of LCLs positive for the HIV marker p24. As depicted in the representative flow cytometry plots in figure 5D the LCL targets contain a mixture of cells: cells that are negative for p24 and cells that are superinfected with HIV (the p24⁺ cells) e.g. 5% of all the potential target cells are EBV/HIV-dual positive the rest are EBV single positive. If in the conditions of co-culture with EBV specific CD8⁺ T cell clones the percentage of p24⁺ cells of all LCLs remained the same this would mean that the clones targeted the HIV⁺ LCLs as frequently as the HIV⁻ LCLs. As such a shift in the percentage is indicative of a preferential targeting of one subset over the other. The HIV negative LCLs are automatically reflected in the percentage minus the HIV positives of the total target cell population.

This is specified in the figure legend as follows:

“Specific elimination of p24⁺ autologous or HLA-matched LCLs by individual EBV-specific CD8⁺ T cell clones expressed as percentage loss of % p24⁺ cells in cocultures with T cell clones compared to conditions without.”

Additionally, from the formula for the calculation of the specific lysis as included in the Methods section it is clear that the ratio EBV/HIV dual positive over EBV single positive cells is considered, not merely the number of HIV infected LCLs:

“Specific lysis of HIV⁺ LCLs was determined by using the formula: % lysis = 100 × (% p24⁺ of PKH67⁺ LCLs without effectors – % p24⁺ of PKH67⁺ LCLs with effectors)/ % p24⁺ PKH67⁺ LCLs without effectors).”

Minor points:

1. Figure 5B should show up and downregulated GO terms for each comparison of data sets not just up for one and down for the other.

There were no downregulated GO terms in HIV-1 infected LCLs compared to non-infected. We have included 3 additional upregulated GO terms for the comparison of HIV-1 infected vs non-infected CD4⁺ T cells in Figure 5B. The full results from the GO term and KEGG analysis are provided to the reader in Supplemental Table S5.

2. The two blue colors on the pie chart in Fig 2A are hard to distinguish so should be changed.

We have changed the color of the undetermined fraction from dark blue to white, which should make it easier to distinguish the characterized integration site categories that are listed in Fig 2A.

3. Coloring and shading in Fig S3D pie charts is very hard to decipher and a better color/shading scheme should be used.

We have changed the coloring and labeling of the pie charts in Fig S3D. We hope this improves the clarity of the presented data.

Reviewer #3 (Comments to the Authors (Required)):

Coinfection of B lymphocytes with EBV and HIV-1 is an important problem for two reasons: first, because HIV-1 increases the risk of EBV-driven B-cell lymphoma, and second because persistence of HIV-1 in another cell type (other than CD4+ T cells) increases the difficulty in eradicating the reservoir of HIV-1 that persists during anti-retroviral treatment.

In the present manuscript, the authors show that X4-tropic HIV-1 (but not R5-tropic HIV-1) can productively infect EBV transformed B cells - lymphoblastoid cell lines (LCLs).

- The HIV provirus is integrated in the B cell genome, with the characteristics typical of HIV infection in T cells, i.e. in active genes, and with a weaker preference for integration in a primary DNA sequence motif.*
- In NSG human mice, there was expansion of the CD8+ T cell population, but not CD4+ T cells, in animals coinfecting with EBV and HIV.*
- EBV-specific CD8+ T cells responded less efficiently with production of IFN γ in the dually-infected humanized mice, and these mice developed macroscopic tumours more frequently than those infected with EBV alone.*
- In apparent contradiction to the last observation, the authors observed that B cells taken from dually-infected mice and transferred to littermates that were reconstituted with the same preparation of human CD34+ cells could transfer HIV-1 infection, but only if the donor had been depleted in vivo of CD8+ T cells before the cells were isolated. Second, the dually-infected B cells were susceptible in vitro to killing by either EBV-specific or HIV-specific CD8+ T cell clones. Thus, although production of IFN γ seemed less efficient, the CD8+ T cells were able to recognize and kill the respective virus-infected cells.*

Comments

The manuscript is clearly written and the results in principle support the main conclusions drawn by the authors. Although infection of B cells with HIV-1 and coinfection of B cells with EBV and HIV-1, have been observed for many years, as the authors acknowledge, the present findings do add to existing knowledge of this coinfection. However, the authors should also acknowledge more clearly some of the significant limitations of the work reported here, as follows.

We thank Reviewer 3 for the expert evaluation of our manuscript and for remarking the potential importance of EBV/HIV-1 co-infection of B cells in relation to lymphomagenesis and as a hurdle for HIV-1 eradication.

Regarding the “apparent contradiction” relating to the last two points of the reviewers summary we would like to point out the following to rule out any possible misconceptions: Figure 3D depicts the EBV specific IFN γ release from splenocytes of individual humanized mice from the respective groups as measured by ELIspot assay. We observed a high EBV specific response in EBV single infected animals and a lower but not absent response in EBV/HIV infected animals. Figure 5D (right) depicts the change in the ratio of EBV single positive and EBV/HIV dual-positive LCLs upon co-culture with EBV specific CD8⁺ T cell clones. In our opinion these experiments do not contradict each other as they address separate issues: In 3D we analyze differences in T cell reactivity from different experimental groups toward a single autologous target (EBV transformed B cells = LCLs). In 5D we analyze whether individual EBV specific CD8⁺ T cells clones, derived from healthy EBV carriers, preferentially eliminate one of two targets (EBV single positive LCLs or EBV/HIV dual positive LCLs) during co-culture. We conclude that due to their higher susceptibility to CD8⁺ T cell recognition EBV/HIV dual-infected B cells are still fairly efficiently cleared in dual-infected mice even so these carry less functional, but not non-functional CD8⁺ T cell populations.

1) *B95-8 has a 10kb deletion in the genome, and is particularly efficient in transforming B cells. Indeed, this is why the strain is the standard strain used to produce LCLs. However, the behavior of the infected B cell differs from that in B cells infected with wild type EBV: the infection leads to initial transformation of the B cell, which then migrates to the germinal centre, where it differentiates to a resting memory state. In contrast, cells infected with B95-8 remain persistently activated. This is likely to change fundamental features of the co-infection with HIV-1, which in consequence are likely to differ from that in cells naturally infected with wild-type EBV and HIV.*

We agree that the B95-8 strain is particularly efficient at transforming B cells *in vitro* for LCL production. It was shown in 2016 by the group around Henri-Jacques Delecluse comparing different EBV strains that B95-8 was among the best B cell transforming whereas the M81 EBV strain, isolated from a Chinese patient with nasopharyngeal carcinoma, was among the least efficient at transforming B cells *in vitro* of the 6 strains studied [28]. Transfer of EBV-infected B cells into NSG mice, however, seemed to lead to similar rates of tumor formation as measured by time until symptom development between the six EBV viruses. M81 virus has an altered tropism and favors epithelial cells with a lower propensity for infecting B cells [28]. Our laboratory teamed up with the afore mentioned group and investigated M81 and B95-8 infection in humanized mice and found that when infectious doses were equalized by ability to infect B cells *in vitro* the M81 virus led to similar or even higher rates of B cell transformation *in vivo*, possibly due to it's higher rate of lytic activity compared to B95-8 [29]. In order to investigate whether HIV-1 infection of LCLs *in vitro* is restricted to the B95-8 strain we infected LCLs derived from M81-infected huNSG mice with NL4-3 HIV-1. We observed an increase of HIV-1 p24 over time in the supernatant of these LCLs derived from different HFL donor reconstituted huNSG mice and present this data in supplemental Figure S1A.

We would also like to point out that EBV strains with large deletions potentially belong to the spectrum of naturally occurring wild type EBV at least in certain pathological scenarios as described recently by Okuno et al. [30]. The authors sequenced the EBV genome from 77 patients with chronic active EBV infection and 61 EBV-associated neoplastic disorders and found intragenic deletions in 27 and 28 cases respectively. The deletions were significantly enriched the BART microRNA clusters (31 cases, Monte Carlo simulation) and frequently affected genes involved in viral particle production (20 cases).

Based on these results and on our results of M81-transformed B cell HIV-1 infection *in vitro* we do not believe that our observations are limited to B95-8. However, we agree with the reviewer that it is difficult to know to what extent the observations are transferable to the dual-infected human population based on our analysis with a focus on B95-8. We have added discussion to this effect as follows:

“In this study we focused our investigations on particular recombinant strains of type 1 EBV (B95-8) and X4-tropic HIV-1 (NL4-3), respectively. It will be necessary to validate these findings using other EBV and HIV-1 strains in future studies to reflect the strain diversity of both viruses in infected human individuals.”

2) *The authors show that HIV-1 infection can persist in the EBV-infected B cells in the humanized mice, although this infection is controlled to an extent by the CD8+ T cell response. However, as the authors are well aware, even without the CD8+ T cell depletion, the humanized mouse model used does not fully and faithfully recapitulate the functioning (and still less the spatial aspects) of the human immune system. Together with the persistent activation of the EBV-transformed B cells studied here, it is therefore difficult to infer what these observations might mean in human infection.*

The reviewer makes an important point here. We have shown that in principal the underlying biology allows for EBV to transform B cells *in vivo* and subsequently for X4-tropic HIV-1 to infect these B cells and that these dual-infected B cells are to an extent controlled by the CD8⁺ T cell response. We have used different *in vitro* and *in vivo* model systems to investigate the potential interaction of these viruses, and as such we think it would be prudent to include text outlining limitations, especially of the humanized mouse model, in the discussion. We have included the following in the discussion:

“Humanized mice have the ability to recapitulate certain aspects of human immune system *in vivo*. However, a number of in limitations in immune function should be mentioned at this point, that may preclude direct transfer of these results to human infection (reviewed in [25]). Importantly, reconstituted human immune system components show similarities to cord blood immune cells. While cell-mediated immune responses can be mounted, the magnitude of these responses may be lower compared to human, isotype switched antibody responses are only rarely observed and steady state levels of IgG are a thousand fold lower than in human serum. [...] Furthermore, this model system can not fully recapitulate the spatioanatomic aspects of human viral infection and lymph nodes and mucosal secondary lymphoid tissues are poorly developed.”

Additional comments

1) *It has recently become clear that Type 2 EBV can infect T cells. It would be interesting to study coinfection of T cells with this wild type EBV and HIV-1.*

We agree that this would be interesting to investigate and have included discussion to this effect:

“Interestingly, type 2 EBV has been described to infect T cells *in vitro*, in humanized mice and in healthy infected children, thus, expanding further the cellular repertoire within which direct interaction of these two important human pathogens could occur [68][69][70].”

2) *It would also be interesting to know whether there is selective infection with HIV-1 of EBV-specific B cells, and finally whether cell-to-cell contact is needed for this infection, i.e. a virological synapse.*

Based on our ability to infect pure cultures of EBV transformed B cells with cell free virus concentrate we do not believe that a virological synapse is needed. Our interpretation of data in Figure 1 and S1 is that direct infection via CD4 and CXCR4 is a sufficient explanation and that CD4 and CXCR4 expression on the surface is the rate-limiting factor. However, along the lines of the reviewer’s query, it is still possible that non-infected LCLs capture the virus to a certain degree on the surface and that they then may make it easier for neighboring cells to get infected. This and any benefits of the B cell receptor specificity for HIV and EBV spreading would be interesting aspects for future studies as EBV infected B cells generate a sizable T

cell response *in vivo* and produce chemotactic factors like CXCL10 that attract T cells to sites of EBV-infected cells (investigated by our laboratory in White et al. [31])

Regarding cell-to-cell transmission, at least of HIV-1 potentially bound to the surface of EBV-infected B cells, we mention the following in the discussion:

“As such, HIV-1 binding on the surface of B cells as immune complexes, as previously described for non-infected B cells [67], may only be poorly modeled in this system. Since EBV infected cells downregulate CD21, the extent to which EBV-infected B cells bind immune-complexed HIV-1 via CD21 and thus facilitate infection of susceptible cells, as described previously for non-EBV infected B cells will have to be further evaluated.”

Additional changes made to comply with formatting guidelines of LSA:

- A summary blurb was added.
- The figure legend titles have been changed in order to not be redundant to Result headers.
- References have been reformatted.
- Additional minor formatting changes have been implemented with no major changes to the text (e.g. figure call-outs).

References

1. Mrozek-Gorska P, Buschle A, Pich D, Schwarzmayr T, Fechtner R, Scialdone A, Hammerschmidt W (2019) Epstein-Barr virus reprograms human B lymphocytes immediately in the prelatent phase of infection *Proc Natl Acad Sci U S A* 116 32:16046-55.
2. Nikitin PA, Yan CM, Forte E, Bocedi A, Tourigny JP, White RE, Allday MJ, Patel A, Dave SS, Kim W, et al. (2010) An ATM/Chk2-mediated DNA damage-responsive signaling pathway suppresses Epstein-Barr virus transformation of primary human B cells *Cell Host Microbe* 8 6:510-22.
3. White RE, Groves IJ, Turro E, Yee J, Kremmer E, Allday MJ (2010) Extensive co-operation between the Epstein-Barr virus EBNA3 proteins in the manipulation of host gene expression and epigenetic chromatin modification *Plos One* 5 11:e13979.
4. Anderton E, Yee J, Smith P, Crook T, White RE, Allday MJ (2008) Two Epstein-Barr virus (EBV) oncoproteins cooperate to repress expression of the proapoptotic tumour-suppressor Bim: clues to the pathogenesis of Burkitt's lymphoma *Oncogene* 27 4:421-33.
5. McHugh D, Caduff N, Barros MHM, Rämer P, Raykova A, Murer A, Landtwing V, Quast I, Styles CT, Spohn M, et al. (2017) Persistent KSHV infection increases EBV-associated tumor formation in vivo via enhanced EBV lytic gene expression *Cell Host & Microbe* 22 1:61-73.
6. Naito T, Tanaka H, Naoe Y, Taniuchi I (2011) Transcriptional control of T-cell development *Int Immunol* 23 11:661-8.
7. He X, He X, Dave VP, Zhang Y, Hua X, Nicolas E, Xu W, Roe BA, Kappes DJ (2005) The zinc finger transcription factor Th-POK regulates CD4 versus CD8 T-cell lineage commitment *Nature* 433 7028:826-33.

8. Pai SY, Truitt ML, Ting CN, Leiden JM, Glimcher LH, Ho IC (2003) Critical roles for transcription factor GATA-3 in thymocyte development *Immunity* 19 6:863-75.
9. Wang L, Wildt KF, Zhu J, Zhang X, Feigenbaum L, Tessarollo L, Paul WE, Fowlkes BJ, Bosselut R (2008) Distinct functions for the transcription factors GATA-3 and ThPOK during intrathymic differentiation of CD4⁺ T cells *Nat Immunol* 9 10:1122-30.
10. Maurice D, Hooper J, Lang G, Weston K (2007) c-Myb regulates lineage choice in developing thymocytes via its target gene *Gata3* *EMBO J* 26 15:3629-40.
11. Caduff N, McHugh D, Murer A, Ramer P, Raykova A, Landtwinig V, Rieble L, Keller CW, Prummer M, Hoffmann L, et al. (2020) Immunosuppressive FK506 treatment leads to more frequent EBV-associated lymphoproliferative disease in humanized mice *PLoS Pathog* 16 4:e1008477.
12. Abbas AR, Baldwin D, Ma Y, Ouyang W, Gurney A, Martin F, Fong S, Campagne MV, Godowski P, Williams PM, et al. (2005) Immune response in silico (IRIS): immune-specific genes identified from a compendium of microarray expression data *Genes Immun* 6 4:319-31.
13. Hoennscheidt C, Max D, Richter N, Staeger MS (2009) Expression of CD4 on Epstein-Barr virus-immortalized B cells *Scand J Immunol* 70 3:216-25.
14. Galli E, Hartmann FJ, Schreiner B, Ingelfinger F, Arvaniti E, Diebold M, Mrdjen D, van der Meer F, Krieg C, Al Nimer F, et al. (2019) GM-CSF and CXCR4 define a T helper cell signature in multiple sclerosis *Nat Med* 25 8:1290-1300.
15. Mei HE, Leipold MD, Schulz AR, Chester C, Maecker HT (2015) Barcoding of Live Human Peripheral Blood Mononuclear Cells for Multiplexed Mass Cytometry *J Immunol* 194 4:2022-31.
16. Smith N, Tierney R, Wei W, Vockerodt M, Murray PG, Woodman CB, Rowe M (2013) Induction of interferon-stimulated genes on the IL-4 response axis by Epstein-Barr virus infected human B cells; relevance to cellular transformation *Plos One* 8 5:e64868.
17. Gras GS, Dormont D (1991) Antibody-dependent and antibody-independent complement-mediated enhancement of human immunodeficiency virus type 1 infection in a human, Epstein-Barr virus-transformed B-lymphocytic cell line *J Virol* 65 1:541-5.
18. Alkhatib G, Combadiere C, Broder CC, Feng Y, Kennedy PE, Murphy PM, Berger EA (1996) CC CKRS: A RANTES, MIP-1 alpha, MIP-1 beta receptor as a fusion cofactor for macrophage-tropic HIV-1 *Science* 272 5270:1955-8.
19. Deng HK, Liu R, Ellmeier W, Choe S, Unutmaz D, Burkhart M, DiMarzio P, Marmon S, Sutton RE, Hill CM, et al. (1996) Identification of a major co-receptor for primary isolates of HIV-1 *Nature* 381 6584:661-6.
20. Dragic T, Litwin V, Allaway GP, Martin SR, Huang YX, Nagashima KA, Cayanan C, Maddon PJ, Koup RA, Moore JP, et al. (1996) HIV-1 entry into CD4⁺ cells is mediated by the chemokine receptor CC-CKR-5 *Nature* 381 6584:667-73.
21. Feng Y, Broder CC, Kennedy PE, Berger EA (1996) HIV-1 entry cofactor: Functional cDNA cloning of a seven-transmembrane, G protein-coupled receptor *Science* 272 5263:872-7.
22. Gras G, Richard Y, Roques P, Olivier R, Dormont D (1993) Complement and virus-specific antibody-dependent infection of normal B lymphocytes by human immunodeficiency virus type 1 *Blood* 81 7:1808-18.
23. Gras G, Legendre C, Krzysiek R, Dormont D, Galanaud P, Richard Y (1996) CD40/CD40L interactions and cytokines regulate HIV replication in B cells in vitro *Virology* 220 2:309-19.
24. Moir S, Lapointe R, Malaspina A, Ostrowski M, Cole CE, Chun TW, Adelsberger J, Baseler M, Hwu P, Fauci AS (1999) CD40-Mediated induction of CD4 and CXCR4 on B lymphocytes correlates with restricted susceptibility to human

immunodeficiency virus type 1 infection: potential role of B lymphocytes as a viral reservoir *J Virol* 73 10:7972-80.

25. Poulin L, Paquette N, Moir S, Lapointe R, Darveau A (1994) Productive infection of normal CD40-activated human B lymphocytes by HIV-1 *Aids* 8 11:1539-44.

26. Blankson JN, Persaud D, Siliciano RF (2002) The challenge of viral reservoirs in HIV-1 infection *Annu Rev Med* 53:557-93.

27. Connor RI, Sheridan KE, Ceradini D, Choe S, Landau NR (1997) Change in coreceptor use correlates with disease progression in HIV-1-infected individuals *J Exp Med* 185 4:621-8.

28. Tsai MH, Lin X, Shumilov A, Bernhardt K, Feederle R, Poirey R, Kopp-Schneider A, Pereira B, Almeida R, Delecluse HJ (2017) The biological properties of different Epstein-Barr virus strains explain their association with various types of cancers *Oncotarget* 8 6:10238-54.

29. Tsai MH, Raykova A, Klinke O, Bernhardt K, Gartner K, Leung CS, Geletneky K, Sertel S, Munz C, Feederle R, et al. (2013) Spontaneous Lytic Replication and Epitheliotropism Define an Epstein-Barr Virus Strain Found in Carcinomas *Cell Reports* 5 2:458-70.

30. Okuno Y, Murata T, Sato Y, Muramatsu H, Ito Y, Watanabe T, Okuno T, Murakami N, Yoshida K, Sawada A, et al. (2019) Defective Epstein-Barr virus in chronic active infection and haematological malignancy (vol 4, pg 404, 2019) *Nat Microbiol* 4 3:544-.

31. White RE, Ramer PC, Naresh KN, Meixlsperger S, Pinaud L, Rooney C, Savoldo B, Coutinho R, Bodor C, Gribben J, et al. (2012) EBNA3B-deficient EBV promotes B cell lymphomagenesis in humanized mice and is found in human tumors *J Clin Invest* 122 4:1487-502.

June 9, 2020

RE: Life Science Alliance Manuscript #LSA-2020-00640-TR

Prof. Christian Munz
University of Zurich
Viral Immunobiology Institute of Experimental Immunology University of Zuerich
Winterthurerstrasse 190
Zuerich, Zurich CH-8057
Switzerland

Dear Dr. Munz,

Thank you for submitting your revised manuscript entitled "EBV renders B cells susceptible to HIV-1 in humanized mice". We would be happy to publish your paper in Life Science Alliance pending final revisions necessary to meet our formatting guidelines.

-please make sure that the author order in the manuscript matches the author order in our system
-please add all mandatory information in our system (the Category is missing)

A. FINAL FILES:

B. MANUSCRIPT ORGANIZATION AND FORMATTING:

Sincerely,

Reilly Lorenz
Editorial Office Life Science Alliance
Meyerhofstr. 1
69117 Heidelberg, Germany
t +49 6221 8891 414
e contact@life-science-alliance.org
www.life-science-alliance.org

Reviewer #1 (Comments to the Authors (Required)):

The authors have thoroughly addressed the concerns that I had raised during the primary review. I have no further comments. Congratulations on this intriguing body of work.

Reviewer #2 (Comments to the Authors (Required)):

The authors have addressed all of my concerns, providing a detailed and very thorough consideration of the points that were raised. They have fully assessed all of the data in the publications I referred to and have examined this in the context of their conclusions and interpretations. They have also sought out further data to confirm their findings. This has strengthened their conclusions for the most part and has led to some revised statements in others.

I am satisfied that the paper describes an important advance and that the data support the conclusions. The requests for more data analysis /statistical testing and presentation improvements have been completed to my satisfaction.

June 11, 2020

RE: Life Science Alliance Manuscript #LSA-2020-00640-TRR

Prof. Christian Munz
University of Zurich
Viral Immunobiology Institute of Experimental Immunology University of Zuerich
Winterthurerstrasse 190
Zuerich, Zurich CH-8057
Switzerland

Dear Dr. Munz,

Thank you for submitting your Research Article entitled "EBV renders B cells susceptible to HIV-1 in humanized mice". It is a pleasure to let you know that your manuscript is now accepted for publication in Life Science Alliance. Congratulations on this interesting work.

*****IMPORTANT:** If you will be unreachable at any time, please provide us with the email address of an alternate author. Failure to respond to routine queries may lead to unavoidable delays in publication.*******

DISTRIBUTION OF MATERIALS:

Again, congratulations on a very nice paper. I hope you found the review process to be constructive and are pleased with how the manuscript was handled editorially. We look forward to future exciting submissions from your lab.

Sincerely,

Reilly Lorenz
Editorial Office Life Science Alliance
Meyerhofstr. 1
69117 Heidelberg, Germany
t +49 6221 8891 414
e contact@life-science-alliance.org
www.life-science-alliance.org